# QuadricFormer: Scene as Superquadrics for 3D Semantic Occupancy Prediction

**Sicheng Zuo**[1,*]  **Wenzhao Zheng**[1,*,†]  **Xiaoyong Han**[1,*]
**Longchao Yang**[2]  **Yong Pan**[2]  **Jiwen Lu**[1,3]

[1]Tsinghua University    [2]Li Auto Inc
[3]Beijing National Research Center for Information Science and Technology
Project Page: https://zuosc19.github.io/QuadricFormer/

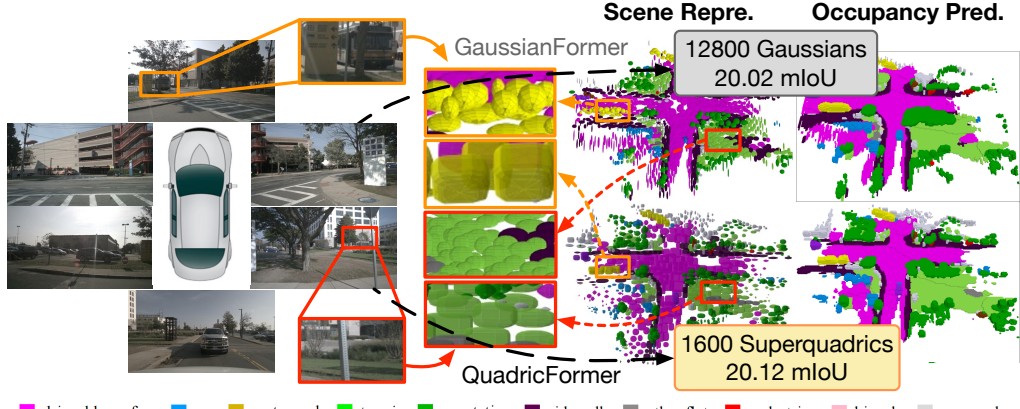

Figure 1: Considering the ellipsoidal shape prior of Gaussians, we propose leveraging expressive superquadrics to build an efficient and powerful object-centric representation. Our QuadricFormer achieves state-of-the-art performance with superior efficiency for 3D occupancy prediction.

## Abstract

3D occupancy prediction is crucial for robust autonomous driving systems as it enables comprehensive perception of environmental structures and semantics. Most existing methods employ dense voxel-based scene representations, ignoring the sparsity of driving scenes and resulting in inefficiency. Recent works explore object-centric representations based on sparse Gaussians, but their ellipsoidal shape prior limits the modeling of diverse structures. In real-world driving scenes, objects exhibit rich geometries (e.g., cuboids, cylinders, and irregular shapes), necessitating excessive ellipsoidal Gaussians densely packed for accurate modeling, which leads to inefficient representations. To address this, we propose to use geometrically expressive superquadrics as scene primitives, enabling efficient representation of complex structures with fewer primitives through their inherent shape diversity. We develop a probabilistic superquadric mixture model, which interprets each superquadric as an occupancy probability distribution with a corresponding geometry prior, and calculates semantics through probabilistic mixture. Building on this, we present QuadricFormer, a superquadric-based model for efficient 3D occupancy prediction, and introduce a pruning-and-splitting module to further enhance modeling efficiency by concentrating superquadrics in occupied regions. Extensive experiments on the nuScenes and KITTI-360 datasets demonstrate that QuadricFormer achieves state-of-the-art performance while maintaining superior efficiency. Code is available at https://github.com/zuosc19/QuadricFormer.

---

*Equal contribution. †Corresponding author.

39th Conference on Neural Information Processing Systems (NeurIPS 2025).

# 1 Introduction

Vision-centric autonomous driving systems have gained much attention for their cost-effectiveness over LiDAR-based solutions [4, 17, 50, 26, 23]. However, they struggle to perceive irregularly shaped obstacles due to visual ambiguity, which compromises driving safety. Recent advances in 3D semantic occupancy prediction address this by estimating voxel-level occupancy status and semantic labels in 3D scenes [43, 44, 38, 39]. This provides a full understanding of scene structures and semantics, which enables applications including self-supervised 3D scene understanding [6, 16], 4D occupancy forecasting [51, 31, 41, 46], and end-to-end autonomous driving [14, 52].

Despite promising applications, 3D semantic occupancy prediction faces efficiency challenges due to its dense 3D predictions [4, 38]. An efficient and expressive 3D representation is therefore essential. While voxel-based methods [23, 44] use dense 3D grids to capture fine details, they ignore the sparsity of driving scenes and suffer from high computational costs. Recent advances introduce object-centric representations using 3D Gaussians [18, 55] to describe scenes sparsely. Each Gaussian models the occupancy probability distribution of its local region via learnable attributes including position, covariance, opacity, and semantics. However, Gaussian representations are fundamentally limited. By their mathematical formulation, Gaussians describe the spatial occupancy probability with an ellipsoidal decay pattern. This imposes a strong ellipsoidal shape prior to Gaussians and severely constrains their capacity to model diverse geometries. Real-world driving scenarios contain objects with rich structural variations, which cannot be accurately represented by a few ellipsoidal Gaussian. Consequently, Gaussian-based models must aggregate numerous densely packed Gaussians to approximate target shapes, causing significant efficiency degradation.

In this paper, we propose an efficient and expressive object-centric 3D representation using superquadrics [1] as scene primitives. Superquadrics are a family of parameterized shapes with high geometric expressiveness and compact shape parameters, offering great flexibility in modeling diverse geometries. This allows superquadrics to model complex structures with sparse packing, enabling an efficient and powerful representation [12]. We represent scenes with a set of learnable superquadrics, each characterized by attributes including position, scale, rotation, opacity, semantics, and shape exponents. For occupancy prediction, we adopt a probabilistic superquadric mixture model that interprets each superquadric as a local occupancy probability distribution, and calculates semantics through probabilistic mixture. Building on this representation, we introduce QuadricFormer, a superquadric-based framework for efficient 3D semantic occupancy prediction. Moreover, we design a pruning-and-splitting module that concentrates superquadrics on occupied regions to further enhance modeling efficiency. Extensive experiments on the nuScenes and KITTI-360 dataset demonstrate that our QuadricFormer achieves state-of-the-art performance with superior efficiency.

# 2 Related Work

**3D Semantic Occupancy Prediction.** 3D semantic occupancy prediction reconstructs fine-grained 3D scenes by labeling each voxel with geometric and semantic information, which is critical for autonomous driving [4, 17, 38, 50, 51]. LiDAR and cameras are the two most commonly used sensors. While LiDAR-based methods excel in depth accuracy [8, 7, 10, 20, 21, 28, 33, 36, 45, 47, 48, 53, 54], their limitations in adverse weather and long-range detection motivate the vision-centric approaches, which reconstruct scenes from multi-view visual input [26, 44, 49, 4, 17]. Early approaches lifted image features directly into dense voxel grids for 3D occupancy prediction [9, 23, 44, 30]. However, given the sparsity of occupied voxels in driving scenes, subsequent works prioritized efficiency through alternative representations. Planar representations like BEV [25] and TPV [17] compress 3D data into 2D feature maps for efficient processing, but sacrifice geometric fidelity. Object-centric modeling preserves geometric fidelity by focusing computation on salient regions [18, 15, 29, 37, 55], alleviating both the redundancy of uniform voxel grids and the information loss from planar compression. However, these methods still struggle to balance efficiency and modeling capacity due to the complexity of real-world structures. To address this, we propose a superquadric-based model that achieves efficient and accurate representation of complex geometries.

**Object-centric Scene Representation.** Existing 3D scene representations primarily use voxel-based frameworks for fine-grained volumetric modeling [44, 23], excelling in semantic prediction tasks. However, their uniform processing of all voxels introduces spatial redundancy, particularly in sparse environments. To address this, recent works explore object-centric representations [37, 29, 18, 15, 55].

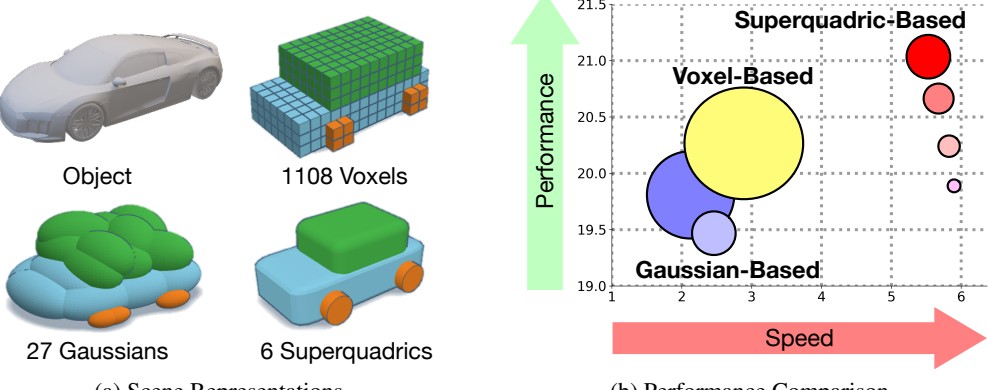

(a) Scene Representations.         (b) Performance Comparison.

Figure 2: **Comparisons between different representations.** (a) Quadric-based method represents the same object with a smaller number of primitives and greater shape expressiveness. (b) Quadric-based representation outperforms existing methods in both accuracy and speed with far fewer primitives.

One line of methods partitions dense grids into localized regions, preserving only detected object areas [37, 29]. While efficient, non-empty regions may be falsely pruned, leading to irreversible loss of critical geometry. Alternatively, point-based methods use sparse points as queries for iterative refinement [34, 40]. However, points inherently lack spatial extent, limiting their ability to capture contextual geometry. Recent advances adopt 3D semantic Gaussians [18, 15, 55], where probability densities radiate from Gaussian centers to enable adaptive spatial coverage. While Gaussians mitigate point rigidity through probability spread, complex geometries often require multiple densely packed primitives, particularly for fine structures, leading to inefficient representations. In this paper, we propose geometrically expressive superquadrics as compact scene primitives. Unlike conventional object-centric methods, superquadrics natively parameterize diverse geometries (e.g., cuboids, cylinders) without dense packing, achieving superior reconstruction fidelity using fewer primitives.

**Superquarics.** Superquadrics are parametric geometric primitives introduced by Barr et al. [1] to model diverse shapes with compact parameterizations. A canonical superquadric is defined by five parameters: three scale parameters along each of its semi-axes and two exponents that determine its shape [19]. The scale and shape parameters of superquadrics allow for smooth interpolation between different geometric shapes, such as cuboids, cylinders, and spheres. When combined with six pose parameters for translation and rotation, a superquadric can represent a complete 3D object using only 11 parameters. Recent works employed superquadrics to decompose complex environments into compact geometric primitives [12]. These methods demonstrate compelling reconstruction capability and editing flexibility, while maintaining model efficiency. However, existing approaches operate exclusively on point clouds and are limited to object-level reconstructions. Differently, we present the first superquadric-based framework for holistic scene reconstruction directly from multi-view images, delivering state-of-the-art performance with superior efficiency.

## 3 Proposed Approach

In this section, we present our method based on the superquadric representation for efficient 3D semantic occupancy prediction. We first review the Gaussian-based object-centric representation and analyze its limitations (Sec 3.1). We then introduce our superquadric representation and probabilistic modeling approach for efficient occupancy prediction (Sec 3.2). Finally, we describe the overall architecture of QuadricFormer for vision-centric 3D occupancy prediction.(Sec 3.3).

### 3.1 Object-Centric Representation

Vision-centric 3D semantic occupancy prediction aims to estimate the occupancy status and semantic label of each voxel in 3D space based on visual inputs. Formally, given input images $\mathcal{I} = \{\mathbf{I}_i\}_{i=1}^{N}$

from $N$ views, the model aims to predict voxel-level semantic labels $\mathbf{O} \in \mathcal{C}^{X \times Y \times Z}$ of the 3D scene, where $\mathcal{C}$ denotes the semantic classes and $X \times Y \times Z$ represents the spatial shape of occupancy.

To achieve this, voxel-based methods [44, 50] adopt dense voxel features to model 3D scenes, resulting in extremely high computational complexity of $\mathcal{O}(XYZ)$. This inefficiency stems from their uniform processing of all voxels in space, which ignores the inherent sparsity of real-world scenes. Considering this, recent works [18, 15] explore object-centric representations based on 3D Gaussians to focus computational resources on salient regions for efficient scene modeling. Gaussian-based method [15] typically employs a set of $P$ semantic 3D Gaussian primitives $\mathcal{G} = \{\mathbf{G}_i\}_{i=1}^P$ to represent 3D scenes sparsely. Each Gaussian $\mathbf{G}_i$ models a flexible local region with its explicit mean $\mathbf{m}_i$, scale $\mathbf{s}_i$, rotation $\mathbf{r}_i$, opacity $a_i$, and semantic probability $\mathbf{c}_i$. For a point $\mathbf{x}$ in 3D space, its geometric occupancy probability associated with the Gaussian $\mathbf{G}$ is:

$$\alpha(\mathbf{x}; \mathbf{G}) = \exp\left(-\frac{1}{2}(\mathbf{x} - \mathbf{m})^{\mathrm{T}} \mathbf{\Sigma}^{-1}(\mathbf{x} - \mathbf{m})\right), \tag{1}$$

$$\mathbf{\Sigma} = \mathbf{R}\mathbf{S}\mathbf{S}^T\mathbf{R}^T, \quad \mathbf{S} = \mathrm{diag}(\mathbf{s}), \quad \mathbf{R} = \mathrm{q2r}(\mathbf{r}), \tag{2}$$

where $\mathbf{x}$ denotes the point position, and $\mathbf{\Sigma}$, $\mathbf{R}$, $\mathbf{S}$ represent the covariance matrix, the rotation matrix constructed from the quaternion $\mathbf{r}$, and the diagonal scale matrix from the scale $\mathbf{s}$. Furthermore, a probabilistic Gaussian mixture model is used to aggregate multiple Gaussians for predicting the structure and semantics of the scene. As each Gaussian represents a flexible region of the scene, the Gaussian-based representation enables adaptive allocation of resources and efficient modeling.

Although 3D Gaussian representation is more efficient than dense voxels (e.g., 6400 Gaussians vs. $200 \times 200 \times 16$ voxels per scene), it still exhibits limitations that prevent an optimal efficiency-performance balance. *Our key insight is that Gaussians inherently impose an ellipsoidal shape prior, which limits their ability to model diverse structures.* As shown in Eq. 1, the occupancy probability distribution of the Gaussian $\mathbf{G}$ can be viewed as a set of iso-probability surfaces defined by:

$$g(\mathbf{x}) = -\frac{1}{2}\left(\left(\frac{x}{s_x}\right)^2 + \left(\frac{y}{s_y}\right)^2 + \left(\frac{z}{s_z}\right)^2\right) = k, \tag{3}$$

where $\mathbf{x} = (x, y, z)^T$ denotes the point position, $k$ denotes the hyperparameter of the surface family, and $\mathbf{s} = (s_x, s_y, s_z)^T$ represents the Gaussian's scales along three axes. The rotation and mean of the Gaussian are omitted for simplicity in Eq. 3, which describes a standard ellipsoid. Each Gaussian then models occupancy probability with an ellipsoidal decay in 3D space. But real-world objects often have diverse shapes, such as cuboids, cylinders, and irregular shapes, which cannot be accurately represented by a few ellipsoidal Gaussians. This forces the model to use numerous densely packed Gaussians to approximate complex structures, leading to inefficient scene representations. In contrast, our method employs expressive superquadrics as scene primitives, enabling efficient and compact modeling of complex structures with only a few sparsely packed superquadrics.

### 3.2 Scene as Superquadrics

We introduce an object-centric scene representation leveraging superquadric primitives for their efficiency and expressive power. Superquadrics are a parametric shape family with strong geometric expressiveness, defined as follows:

$$f(\mathbf{x}) = \left(\left(\frac{x}{s_x}\right)^{\frac{2}{\epsilon_2}} + \left(\frac{y}{s_y}\right)^{\frac{2}{\epsilon_2}}\right)^{\frac{\epsilon_2}{\epsilon_1}} + \left(\frac{z}{s_z}\right)^{\frac{2}{\epsilon_2}} = k, \tag{4}$$

where $\mathbf{x} = (x, y, z)^T$ denotes the point position, and $k$ denotes the hyperparameter of the surface family. Compared to the ellipsoids in Eq. 3, superquadrics introduce only two additional shape-defining exponents $\epsilon_1, \epsilon_2$ yet can represent a much wider variety of shapes. As shown in Fig. 2a, superquadrics allow for continuous and diverse shape variations as the shape parameters change. This inherent parameter efficiency and geometric expressiveness enable superquadrics to model diverse shapes without being densely packed. Consequently, only a small number of superquadrics are needed to represent complex scene structures, achieving an efficient yet powerful scene representation.

We thus utilize a set of $P$ parameterized superquadrics $\mathcal{Q} = \{\mathbf{Q}_i\}_{i=1}^P$ to represent the 3D scene. Each superquadric is characterized by its scale $\mathbf{s}$ and shape exponents $\epsilon_1, \epsilon_2$ to define its geometry. To extend the representation to the global coordinate system, each primitive is also assigned a

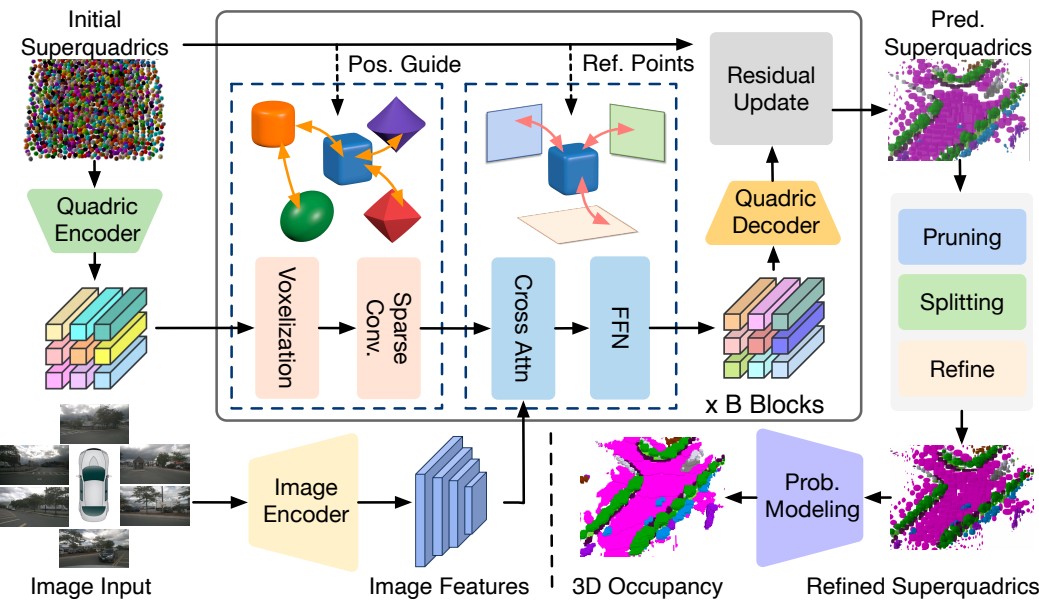

Figure 3: **Overall Framework of QuadricFormer.** We use several quadric-encoder blocks to update superquadrics, and employ a pruning-and-splitting module to further enhance modeling efficiency.

position $\mathbf{x}$ and rotation $\mathbf{r}$. Beyond geometric attributes, each superquadric is further equipped with an opacity $a$ and a semantic probability $\mathbf{c}$ to incorporate semantic information. In summary, our superquadric-based representation can be formulated as:

$$\mathcal{Q} = \{\mathbf{Q}_i\}_{i=1}^P = \{(\mathbf{x}_i, \mathbf{s}_i, \mathbf{r}_i, a_i, \epsilon_{1;i}, \epsilon_{2;i}, \mathbf{c}_i,)\}_{i=1}^P. \tag{5}$$

We now explore obtaining 3D occupancy prediction from the superquadric representation. Existing methods [12] typically treat superquadrics as deterministic surfaces, fitting them to object parts for point cloud reconstruction. However, these surface-based approaches face key limitations in vision-centric occupancy prediction. A primary challenge is supervision. While point cloud reconstruction directly optimizes the distance between points and the superquadric surfaces, occupancy prediction requires fine-grained scene understanding, which lacks clear surface-based constraints. Furthermore, surface-based methods rely on the explicit structure from point cloud inputs, whereas visual inputs introduce structural uncertainty, making deterministic modeling unstable. Lastly, surface-based methods focus on object-level reconstruction with simple spatial relationships. But real-world driving scenes involve far more complex surface interactions, posing significant modeling difficulties.

To achieve robust 3D semantic occupancy prediction, we design a probabilistic modeling mechanism that converts superquadrics into occupancy probabilities. Inspired by GaussianFormer-2 [15], we adopt a probabilistic superquadric mixture model, where each superquadric defines the occupancy probability distribution in its local neighborhood. To compute the probability of a 3D point $\mathbf{x}$ being occupied by the superquadric $\mathbf{Q}$, we first transform $\mathbf{x}$ into $\mathbf{Q}$'s local coordinate system, defined by its position $\mathbf{m}$ and rotation $\mathbf{r}$:

$$\mathbf{x_Q} = \mathbf{R}(\mathbf{x} - \mathbf{m}), \tag{6}$$

where $\mathbf{x_Q}$ denotes the local coordinate of $\mathbf{x}$, and $\mathbf{R}$ denotes the rotation matrix constructed from the rotation $\mathbf{r}$. The occupancy probability of $\mathbf{x}$ associated with $\mathbf{Q}$ is then computed as:

$$p_{\mathbf{o}}(\mathbf{x}; \mathbf{G}) = \exp\big(-\mathrm{f}(\mathbf{x_Q})\big) = \exp\left(\left(\left(\frac{x_\mathbf{Q}}{s_x}\right)^{\frac{2}{\epsilon_2}} + \left(\frac{y_\mathbf{Q}}{s_y}\right)^{\frac{2}{\epsilon_2}}\right)^{\frac{\epsilon_2}{\epsilon_1}} + \left(\frac{z_\mathbf{Q}}{s_z}\right)^{\frac{2}{\epsilon_2}}\right), \tag{7}$$

where $\mathbf{x_Q} = (x_\mathbf{Q}, y_\mathbf{Q}, z_\mathbf{Q})^T$ and $\mathbf{s} = (s_x, y_x, z_s)^T$ are the position and scale parameters, respectively, and $\epsilon_1, \epsilon_2$ are the shape exponents of the superquadric $\mathbf{Q}$. Assuming conditional independence of occupancy among different superquadrics, the final occupancy probability at $\mathbf{x}$ is computed as:

$$p_{\mathbf{o}}(\mathbf{x}) = 1 - \prod_{i=1}^P \big(1 - p_{\mathbf{o}}(\mathbf{x}; \mathbf{Q}_i)\big). \tag{8}$$

Semantic predictions are subsequently inferred by a weighted aggregation of semantic probabilities from all contributing superquadrics, where weights correspond to their occupancy influence at $\mathbf{x}$:

$$p_{\mathbf{c}}(\mathbf{x}) = \frac{\sum_{i=1}^{P} p_{\mathbf{o}}(\mathbf{x}|\mathbf{Q}_i)a_i\mathbf{c}_i}{\sum_{j=1}^{P} p_{\mathbf{o}}(\mathbf{x}|\mathbf{Q}_j)a_j}, \tag{9}$$

The key to this probabilistic modeling is incorporating the superquadric geometry as shape priors within the probability distribution, realized as iso-probability surfaces conforming to its geometry in Eq. 4. Leveraging the geometrically expressive power of superquadrics, our model can efficiently represent complex 3D structures using a sparse set of primitives without dense packing, achieving an efficient yet powerful scene representation. Moreover, this probabilistic framework effectively models structural uncertainties arising from visual ambiguities, significantly improving the model's robustness and generalization capabilities.

### 3.3 QuadricFormer

We present the overall framework of QuadricFormer in Fig. 3. Starting from the image inputs of $N$ views $\mathcal{I} = \{\mathbf{I}_i\}_{i=1}^{N}$, we first employ an image backbone $E_{\mathbf{I}}$ to extract multi-scale image features $\mathbf{F}_{\mathbf{I}}$:

$$\mathbf{F}_{\mathbf{I}} = E_{\mathbf{I}}(\mathcal{I}), \tag{10}$$

Due to the lack of any structural prior of the scene, we randomly initialize a few superquadrics $\mathbf{Q}_{init}$ in 3D space, and use $B$ quadric-encoder blocks $E_{\mathbf{B}}$ to predict the final superquadrics from images. In each block, we first encode current superquadrics $\mathbf{Q}_i$ into features $\mathbf{F}_{\mathbf{Q}}$ via a quadric encoder $E_{\mathbf{Q}}$:

$$\mathbf{F}_{\mathbf{Q}} = E_{\mathbf{Q}}(\mathbf{Q}_i). \tag{11}$$

We then use 3D sparse convolution $E_{conv}$ for superquadric feature self-encoding and deformable attention $E_{attn}$ for interaction between superquadric and image features:

$$\mathbf{F}_{\mathbf{Q}} = E_{conv}(\mathbf{F}_{\mathbf{Q}}, \mathbf{x}_{\mathbf{Q}}), \mathbf{F}_{\mathbf{Q}} = E_{attn}(\mathbf{F}_{\mathbf{Q}}, \mathbf{x}_{\mathbf{Q}}, \mathbf{F}_{\mathbf{I}}), \tag{12}$$

where $\mathbf{x}_{\mathbf{Q}}$ denotes the explicit position of the superquadric $\mathbf{Q}$, serving as auxiliary information to guide feature encoding. Finally, a quadric decoder $D_{\mathbf{Q}}$ is used to predict the update of superquadric attributes $\Delta\mathbf{Q}$, which are combined with the original attributes $\mathbf{Q}$ via residual addition:

$$\Delta\mathbf{Q} = D_{\mathbf{Q}}(\mathbf{F}_{\mathbf{Q}}), \mathbf{Q}_{i+1} = \mathbf{Q}_i + \Delta\mathbf{Q}. \tag{13}$$

After $B$ blocks update, we get the final superquadric prediction $\mathbf{Q}$, and the 3D semantic occupancy prediction $\mathbf{O} \in \mathcal{C}^{X \times Y \times Z}$ can be inferred through the probabilistic modeling mechanism:

$$\mathbf{O} = Prob(\mathbf{Q}). \tag{14}$$

For optimization, we adopt the cross entropy loss and the lovaszsoftmax [2] loss for training.

Due to the lack of structural priors, superquadrics are uniformly initialized in 3D space. As a result, some superquadrics in empty regions are optimized to small scales and contribute little to scene modeling, which leads to inefficiency. To address this, we introduce a pruning-splitting module after initial training. Small-scale superquadrics (likely in empty regions) are pruned, while large-scale ones (likely in occupied regions) are split for finer modeling. We keep the number of superquadrics unchanged and use two additional blocks to further refine their properties. Notably, this lightweight module improves superquadric utilization for more efficient scene representation without introducing significant computational overhead.

## 4 Experiments

### 4.1 Datasets and Metrics

**nuScenes** [3] comprises 1,000 urban driving sequences collected in Boston and Singapore. The dataset is officially split into 700 sequences for training, 150 for validation, and 150 for testing. Each sequence spans a duration of 20 seconds with RGB images captured by 6 surrounding cameras, and the key frames are annotated at a 2 Hz frequency. For supervision and evaluation, we leverage the dense semantic occupancy annotations from SurroundOcc. The annotated voxel grid extends from

Table 1: **3D semantic occupancy prediction results on nuScenes.** * means supervised by dense occupancy annotations as opposed to original LiDAR segmentation labels. Sq. denotes the number of Superquadrics in our model. Our method achieves state-of-the-art performance.

| Method | IoU | mIoU | barrier | bicycle | bus | car | const. veh. | motorcycle | pedestrian | traffic cone | trailer | truck | drive. suf. | other flat | sidewalk | terrain | manmade | vegetation |
|---|---|---|---|---|---|---|---|---|---|---|---|---|---|---|---|---|---|---|
| MonoScene [5] | 23.96 | 7.31 | 4.03 | 0.35 | 8.00 | 8.04 | 2.90 | 0.28 | 1.16 | 0.67 | 4.01 | 4.35 | 27.72 | 5.20 | 15.13 | 11.29 | 9.03 | 14.86 |
| Atlas [32] | 28.66 | 15.00 | 10.64 | 5.68 | 19.66 | 24.94 | 8.90 | 8.84 | 6.47 | 3.28 | 10.42 | 16.21 | 34.86 | 15.46 | 21.89 | 20.95 | 11.21 | 20.54 |
| BEVFormer [25] | 30.50 | 16.75 | 14.22 | 6.58 | 23.46 | 28.28 | 8.66 | 10.77 | 6.64 | 4.05 | 11.20 | 17.78 | 37.28 | 18.00 | 22.88 | 22.17 | 13.80 | **22.21** |
| TPVFormer [17] | 11.51 | 11.66 | 16.14 | 7.17 | 22.63 | 17.13 | 8.83 | 11.39 | 10.46 | 8.23 | 9.43 | 17.02 | 8.07 | 13.64 | 13.85 | 10.34 | 4.90 | 7.37 |
| TPVFormer* [17] | 30.86 | 17.10 | 15.96 | 5.31 | 23.86 | 27.32 | 9.79 | 8.74 | 7.09 | 5.20 | 10.97 | 19.22 | 38.87 | 21.25 | 24.26 | 23.15 | 11.73 | 20.81 |
| OccFormer [50] | 31.39 | 19.03 | 18.65 | 10.41 | 23.92 | 30.29 | 10.31 | 14.19 | 13.59 | 10.13 | 12.49 | 20.77 | 38.78 | 19.79 | 24.19 | 22.21 | 13.48 | 21.35 |
| SurroundOcc [44] | 31.49 | 20.30 | 20.59 | 11.68 | 28.06 | 30.86 | 10.70 | 15.14 | **14.09** | **12.06** | 14.38 | 22.26 | 37.29 | 23.70 | 24.49 | 22.77 | **14.89** | 21.86 |
| GaussianFormer [18] | 29.83 | 19.10 | 19.52 | 11.26 | 26.11 | 29.78 | 10.47 | 13.83 | 12.58 | 8.67 | 12.74 | 21.57 | 39.63 | 23.28 | 24.46 | 22.99 | 9.59 | 19.12 |
| GaussianFormer-2 [15] | 31.74 | 20.82 | **21.39** | **13.44** | **28.49** | 30.82 | 10.92 | 15.84 | 13.55 | 10.53 | 14.04 | **22.92** | 40.61 | 24.36 | 26.08 | 24.27 | 13.83 | 21.98 |
| **QuadricFormer (1600 Sq.)** | 31.22 | 20.12 | 19.58 | 13.11 | 27.27 | 29.64 | 11.25 | 16.26 | 12.65 | 9.15 | 12.51 | 21.24 | 40.20 | 24.34 | 25.69 | 24.24 | 12.95 | 21.86 |
| **QuadricFormer (12800 Sq.)** | **32.13** | **21.11** | 21.38 | 13.41 | 28.40 | **31.01** | **11.32** | **17.10** | 13.94 | 11.28 | **14.75** | 22.66 | **40.81** | **24.71** | **26.51** | **25.22** | 13.54 | 21.78 |

Table 2: **Monocular 3D semantic occupancy prediction results on SSCBench-KITTI-360.** Num of Prims. denotes the number of primitives in the model. Our method achieves comparable performance to GaussianFormer-2 [15] with much fewer primitives.

| Method | Input | Num of Prims. | IoU | mIoU | car | bicycle | motorcycle | truck | other-veh. | person | road | parking | sidewalk | other-grnd | building | fence | vegetation | terrain | pole | traf.-sign | other-struct. | other-object |
|---|---|---|---|---|---|---|---|---|---|---|---|---|---|---|---|---|---|---|---|---|---|---|
| LMSCNet [33] | L | - | 47.53 | 13.65 | 20.91 | 0 | 0 | 0.26 | 0 | 0 | 62.95 | 13.51 | 33.51 | 0.2 | 43.67 | 0.33 | 40.01 | 26.80 | 0 | 0 | 3.63 | 0 |
| SSCNet [35] | L | - | 53.58 | 16.95 | 31.95 | 0 | 0.17 | 10.29 | 0.58 | 0.07 | 65.7 | 17.33 | 41.24 | 3.22 | 44.41 | 6.77 | 43.72 | 28.87 | 0.78 | 0.75 | 8.60 | 0.67 |
| MonoScene [5] | C | 262144 | 37.87 | 12.31 | 19.34 | 0.43 | 0.58 | 8.02 | 2.03 | 0.86 | 48.35 | 11.38 | 28.13 | 3.22 | 32.89 | 3.53 | 26.15 | 16.75 | 6.92 | 5.67 | 4.20 | 3.09 |
| Voxformer [24] | C | 262144 | 38.76 | 11.91 | 17.84 | 1.16 | 0.89 | 4.56 | 2.06 | 1.63 | 47.01 | 9.67 | 27.21 | 2.89 | 31.18 | 4.97 | 28.99 | 14.69 | 6.51 | 6.92 | 3.79 | 2.43 |
| TPVFormer [17] | C | 81920 | 40.22 | 13.64 | 21.56 | 1.09 | 1.37 | 8.06 | 2.57 | 2.38 | 52.99 | 11.99 | 31.07 | 3.78 | 34.83 | 4.80 | 30.08 | 17.51 | 7.46 | 5.86 | 5.48 | 2.70 |
| OccFormer [50] | C | 262144 | **40.27** | 13.81 | **22.58** | 0.66 | 0.26 | 9.89 | 3.82 | 2.77 | **54.30** | 13.44 | 31.53 | 3.55 | **36.42** | 4.80 | 31.00 | **19.51** | **7.77** | **8.51** | 6.95 | 4.60 |
| GaussianFormer [18] | C | 38400 | 35.38 | 12.92 | 18.93 | 1.02 | **4.62** | **18.07** | 7.59 | 3.35 | 45.47 | 10.89 | 25.03 | **5.32** | 28.44 | 5.68 | 29.54 | 8.62 | 2.99 | 2.32 | 9.51 | **5.14** |
| GaussianFormer-2 [15] | C | 38400 | 38.37 | **13.90** | 21.08 | **2.55** | 4.21 | 12.41 | 5.73 | 1.59 | 54.12 | 11.04 | **32.31** | 3.34 | 32.01 | 4.98 | 28.94 | 17.33 | 3.57 | 5.48 | 5.88 | 3.54 |
| **QuadricFormer** | C | 12800 | 38.89 | 13.63 | 18.80 | 1.31 | 4.43 | 16.57 | **8.57** | **3.44** | 45.49 | **13.77** | 25.92 | 5.25 | 29.73 | **6.73** | **31.95** | 9.13 | 4.47 | 4.02 | **10.88** | 4.88 |

-50m to 50m along both the X and Y axes, and from -5m to 3m along the Z axis, with a spatial resolution of $200 \times 200 \times 16$. Each voxel is classified into one of the 18 categories(16 semantics, 1 empty and 1 unknown).

**KITTI-360** [27] comprises over 320k images collected in suburban driving scenes with comprehensive 360° sensory coverage, including two perspective cameras, two fisheye cameras, a Velodyne LiDAR, and a laser scanner. In our experiments, we use the RGB images from the left perspective camera of the ego vehicle as model input. For supervision and evaluation, we adopt the semantic occupancy annotations provided by SSCBench-KITTI-360 [22]. The official split contains 7 sequences for training, 1 for validation, and 1 for testing, corresponding to 8487, 1812, and 2566 key frames, respectively. The annotated voxel grid covers a region of $51.2 \times 51.2 \times 6.4$ m$^3$ in front of the ego vehicle, with a spatial resolution of $256 \times 256 \times 32$. Each voxel is categorized into one of 19 classes (18 semantic classes and 1 empty).

**The evaluation metrics** adhere to common practice, namely mean Intersection-over-Union (mIoU) and Intersection-over-Union (IoU):

$$\mathbf{mIoU} = \frac{1}{|\mathcal{C}'|} \sum_{i \in \mathcal{C}'} \frac{TP_i}{TP_i + FP_i + FN_i}, \tag{15}$$

$$\mathbf{IoU} = \frac{TP_{\neq c_0}}{TP_{\neq c_0} + FP_{\neq c_0} + FN_{\neq c_0}}, \tag{16}$$

Where $\mathcal{C}'$, $c_0$, TP, FP, and FN represent the non-empty classes, the empty class, and the number of true positive, false positive, and false negative predictions, respectively.

## 4.2 Implementation Details

The input images are at resolutions of 900×1600 for nuScenes and 376×1408 for KITTI-360 [27] with random flipping and photometric distortion augmentations. We employ ResNet101-DCN [13] with FCOS3D checkpoint [42] for nuScenes [3], and ResNet50 [13] pretrained on ImageNet [11]

Table 3: **Performance and efficiency comparison with Gaussian-based methods.** The latency and memory are tested on an NVIDIA 4090 GPU with batch size one during inference, in accordance with Gaussian-based methods [18, 15]. Our method achieves better performance-efficiency trade-off.

| Method | Number of Primitives | Latency (ms) | Memory (MB) | mIoU | IoU |
|---|---|---|---|---|---|
| GaussianFormer [18] | 25600 | 227 | 4850 | 16.00 | 28.72 |
|  | 144000 | 372 | 6229 | 19.10 | 29.83 |
| GaussianFormer-2 [15] (Depth Initialized) | 1600 | 341 | 3075 | 18.73 | 28.99 |
|  | 3200 | 355 | 3076 | 18.75 | 29.64 |
|  | 6400 | 395 | 3652 | 19.55 | 30.37 |
|  | 12800 | 451 | 4535 | 19.69 | 30.43 |
| QuadricFormer (Ours) | 1600 | **162** | **2554** | 20.04 | 30.71 |
|  | 3200 | 164 | 2556 | 20.35 | 31.62 |
|  | 6400 | 165 | 2560 | 20.79 | 31.89 |
|  | 12800 | 179 | 2563 | **21.11** | **32.13** |

for KITTI-360 [27]. The numbers of Superquarics are set to 1600 in our main results for nuScenes and KITTI-360. For optimization, we train our model using AdamW with weight decay of 0.01, and maximum learning rate of $4 \times 10^{-4}$, which decays with a cosine schedule. We train our model for 20 epochs on nuScenes and KITTI-360 with a batch of 8.

## 4.3 Main Results

**3D Semantic Occupancy Prediction.** We report the performance of our QuadricFormer on nuScenes dataset [3] in Table 1. Compared to other methods, our approach achieves state-of-the-art performance. Specifically, QuadricFormer outperforms other methods on categories such as bicycle, motorcycle, truck and various ground-related classes (drivable surface, sidewalk, terrain, etc.), demonstrating superior capability in modeling both small and structural objects. Moreover, our method significantly surpasses GaussianFormer-2 [15] while using substantially fewer superquadrics (1600 vs. 12800), further validating its efficiency and effectiveness. Furthermore, We report the results for monocular 3D semantic occupancy prediction on SSCBench-KITTI-360 [22] in Table 2. Our method achieves comparable mIoU performance to GaussianFormer-2 [15], demonstrating the effectiveness of our approach for monocular 3D semantic occupancy prediction.

**Performance and Efficiency Comparison with Gaussian-based Methods.** We report the performance and efficiency comparison for QuadricFormer with Gaussian-based methods on nuScenes in Table 3. QuadricFormer consistently outperforms prior methods in both 3D semantic occupancy prediction and computational efficiency. Specifically, our method achieves the highest mIoU (up to 21.11) and IoU (up to 32.13), surpassing all Gaussian-based approaches. In terms of efficiency, QuadricFormer significantly reduces both latency and memory usage. For similar or even fewer primitives (e.g., 1600 or 3200), our method achieves a latency as low as 162 ms and 2554 MB memory consumption, which are substantially lower than others. Notably, even when increasing the number of primitives in QuadricFormer to 12800, both latency and memory usage remain lower than those of Gaussian-based methods using only 1600 primitives. This further highlights the superior efficiency of our approach for the complex structures in real-world applications.

## 4.4 Ablation Study

**Effect of the $\epsilon$ Range.** We conduct ablation study on the range of the superquadric exponent parameters $\epsilon$ in Eq. 4, as reported in Table 4. We set the number of superquadrics to 12800 for these experiments. The table explores the effect of different $\epsilon$ ranges on 3D semantic occupancy prediction performance. We observe that setting the range of $(0.1, 2)$ yields the best results, achieving the highest mIoU (20.51) and IoU (31.25).

**Effect of the Pruning-splitting Module.** We conduct ablation studies on the effect of the pruning-splitting module, as shown in Table 5. The results demonstrate that increasing the crop & split number consistently improves performance. This confirms that reallocating primitives from low to high occupancy regions effectively enhances the accuracy and efficiency of our 3D scene representation.

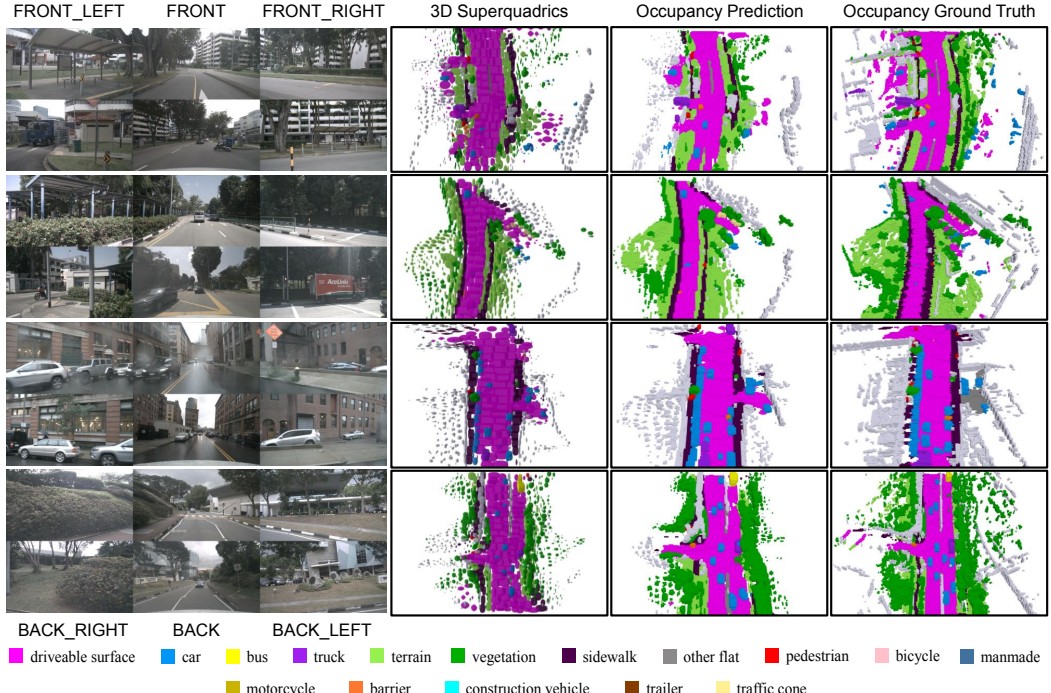

| FRONT_LEFT | FRONT | FRONT_RIGHT | 3D Superquadrics | Occupancy Prediction | Occupancy Ground Truth |

BACK_RIGHT    BACK    BACK_LEFT

■ driveable surface   ■ car   ■ bus   ■ truck   ■ terrain   ■ vegetation   ■ sidewalk   ■ other flat   ■ pedestrian   ■ bicycle   ■ manmade

■ motorcycle   ■ barrier   ■ construction vehicle   ■ trailer   ■ traffic cone

Figure 4: **3D Superquadrics and occupancy visualizations on nuScenes.** Our model is able to predict high-fidelity shapes and achieves comprehensive occupancy results.

Table 4: Effect of the $\epsilon$ range.

| Range of $\epsilon$ | mIoU | IoU |
|---|---|---|
| $(0.01, 2)$ | 20.39 | 31.13 |
| $(0.01, 5)$ | 20.25 | 30.63 |
| $(0.1, 2)$ | **20.51** | **31.25** |
| $(0.1, 5)$ | 19.86 | 30.65 |

Table 5: Effect of the pruning-splitting module.

| Crop & Split Number | mIoU | IoU |
|---|---|---|
| 0 | 19.41 | 39.77 |
| 200 | 19.65 | 30.35 |
| 400 | 19.90 | 30.67 |
| 800 | **20.12** | **31.22** |

## 4.5 Visualizations

We present visualizations of the predicted superquadrics and occupancy results in Figure 4. Our model is able to predict high-fidelity shapes using superquadrics and achieves comprehensive occupancy results. Further, we compare our method against GaussianFormer-2 [15] in Figure 5, showing that our predicted superquadrics offer more adaptive shapes than Gaussians. Moreover, our method achieves high-quality performance using only 1600 superquadrics, compared to 6400 Gaussians. Figure 6 shows a sample for 3D semantic occupancy prediction on the nuScenes [3] validation set. Compared to GaussianFormer-2 [15], our QuadricFormer exhibits enhanced modeling capability for complex objects and road surfaces.

## 5   Conclusion

In this paper, we have proposed a superquadric-based object-centric representation for efficient 3D semantic occupancy prediction. Specifically, we leverage the geometric expressiveness of superquadrics to model complex structures with far fewer sparsely packed primitives. We formulate a probabilistic superquadric mixture model, where each superquadric encodes an occupancy probability distribution with a corresponding geometry prior, and semantics are inferred via probabilistic mixture. Furthermore, we introduce a pruning-and-splitting module that adaptively concentrates superquadrics in occupied regions to further enhance modeling efficiency. Our proposed QuadricFormer demonstrates state-of-the-art performance and superior efficiency on the nuScenes benchmark, providing an effective and compact solution for scene understanding in vision-centric autonomous driving systems.

**Limitations.** With random initialization, QuadricFormer cannot fully learn accurate superquadric positions, leaving some superquadrics in empty regions and reducing representation efficiency.

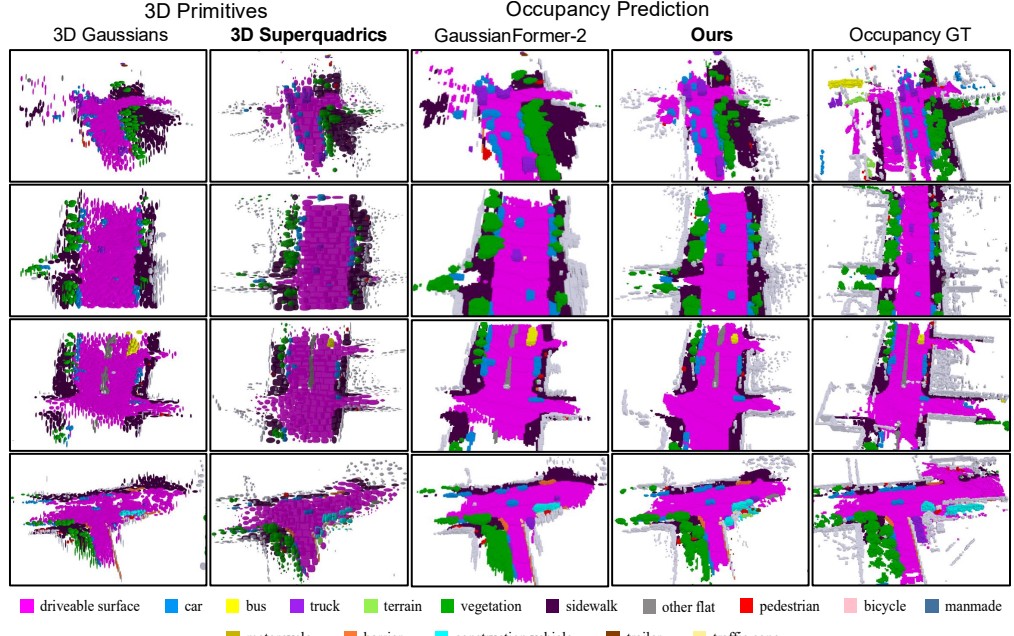

3D Primitives: 3D Gaussians, **3D Superquadrics**

Occupancy Prediction: GaussianFormer-2, **Ours**, Occupancy GT

Legend: driveable surface, car, bus, truck, terrain, vegetation, sidewalk, other flat, pedestrian, bicycle, manmade, motorcycle, barrier, construction vehicle, trailer, traffic cone

Figure 5: **Qualitative comparisons.** QuadricFormer predicts more flexible and adaptive shapes.

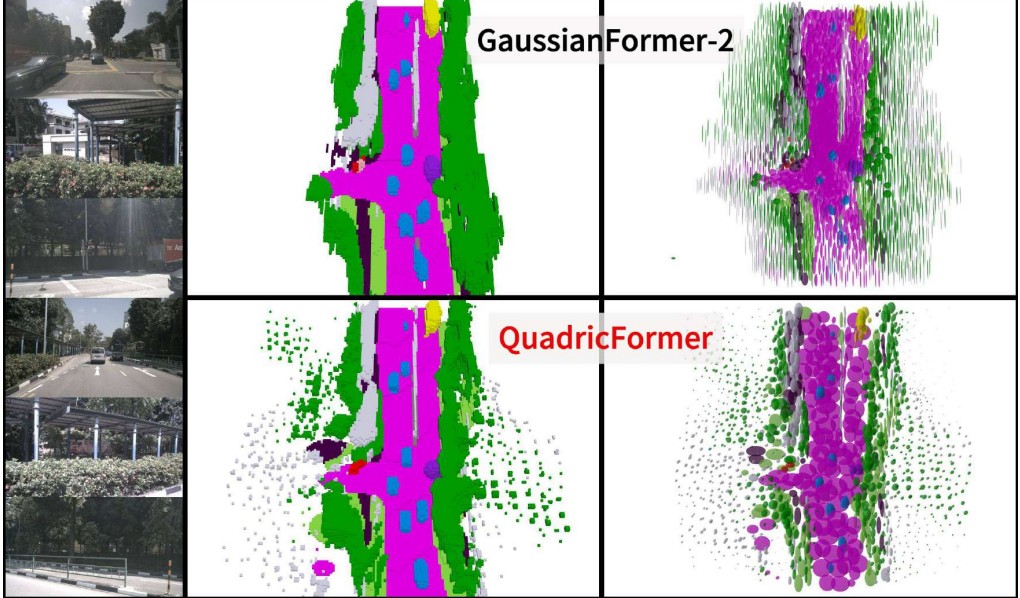

Figure 6: **Visualizations of the proposed QuadricFormer compared to GaussianFormer-2 [15] for 3D semantic occupancy prediction on the nuScenes [3] validation set.** We visualize the six surrounding camera inputs, the corresponding occupancy prediction results, and the primitive representations. The upper row shows the predicted occupancy (left) and the primitive representation (right) by GaussianFormer-2. The lower row shows the prediction results of QuadricFormer.

**Broader impact.** Our work on autonomous driving has the potential to improve traffic efficiency in the future, but it may also contribute to job displacement for drivers.

# Acknowledgements

This work was supported in part by the National Natural Science Foundation of China under Grant 62576188, Grant 62336004, Grant 62321005, and Grant 62125603, in part by the Beijing Natural Science Foundation under Grant L247009, and in part by the Beijing National Research Center for Information Science and Technology.

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

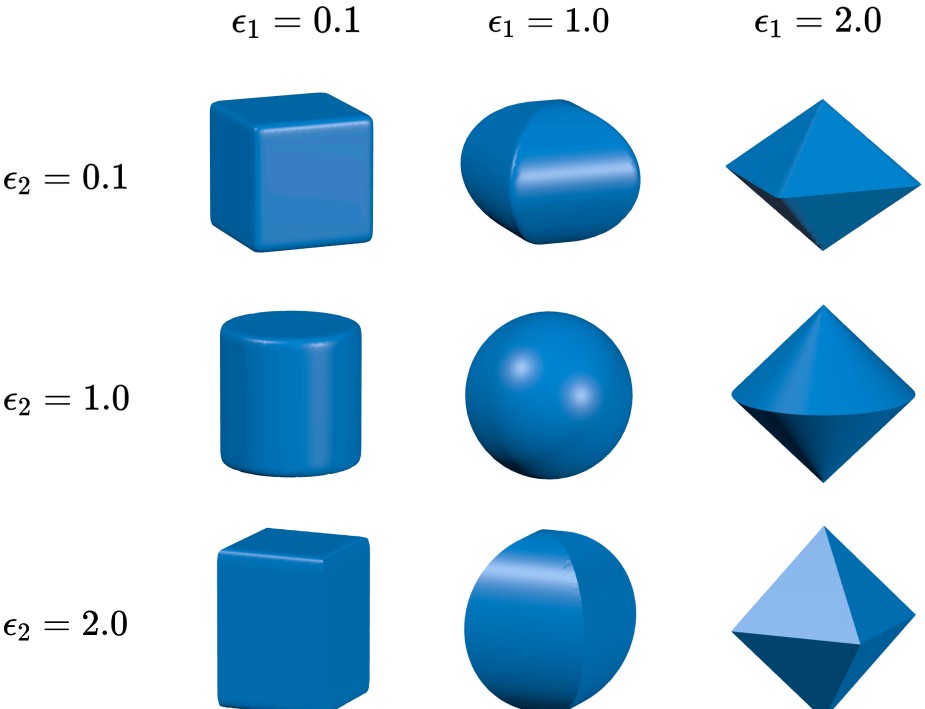

$\epsilon_1 = 0.1$ $\epsilon_1 = 1.0$ $\epsilon_1 = 2.0$

$\epsilon_2 = 0.1$

$\epsilon_2 = 1.0$

$\epsilon_2 = 2.0$

Figure 7: **Superquadrics of different shape parameters.** The figure illustrates how varying $\epsilon_1$ and $\epsilon_2$ produces a wide range of shapes, from star-like and rounded shapes to square-like structures. Such diversity enables superquadrics to flexibly model complex object geometries in 3D scenes.

## A    Additional Superquadric Details

Superquadrics are a powerful family of parameterized surfaces that can represent various geometric shapes. With just a few parameters, superquadrics can generate shapes ranging from basic ellipsoids, cuboids, and cylinders to more complex shapes with rounded corners, star-like profiles, and smooth transitions between them. This geometric flexibility makes superquadrics ideal for efficiently modeling diverse objects in autonomous driving scenes. The shape of a superquadric is mainly controlled by two groups of parameters. The first group consists of scaling factors $(s_x, s_y, s_z)$, which define the superquadric's dimensions or "radii" along its three principal axes, determining the object's overall size and aspect ratio. The second group includes two key shape parameters $(\epsilon_1, \epsilon_2)$ that determine the degree of "squareness" or "roundness" of the object. $\epsilon_1$ primarily controls the object's profile in planes containing the z-axis (such as the xz- or yz-plane): smaller values (close to 0.1) create sharper profiles, $\epsilon_1 = 1.0$ produces elliptical outlines, and larger values (up to 2.0) result in flatter contours. Similarly, $\epsilon_2$ controls the shape of the cross-section in the xy-plane. A small $\epsilon_2$ yields a star-shaped cross-section, $\epsilon_2$=1.0 gives a circular outline, and large $\epsilon_2$ values produce square-like shapes. As shown in Fig 7, varying $\epsilon_1$ and $\epsilon_2$ of superquadrics results in a wide range of shapes. By combining these scaling and shape parameters, superquadrics can efficiently represent diverse object geometries in autonomous driving scenes. This capability allows them to capture complex structures with significantly fewer primitives than traditional representations (like ellipsoidal Gaussians), highlighting their superior modeling efficiency and expressive power for 3D scene understanding tasks.

## B    Additional Experiments

We visualize the position distributions of scene primitives using 1600 superquadrics versus 6400 Gaussians in Figure 8. Gaussian-based methods require a dense arrangement of Gaussians throughout the entire 3D space to model the scene, leading to numerous redundant Gaussians and low modeling efficiency. In contrast, our superquadric-based method learns well-structured spatial arrangements, enabling it to effectively model the scene structure with significantly fewer primitives.

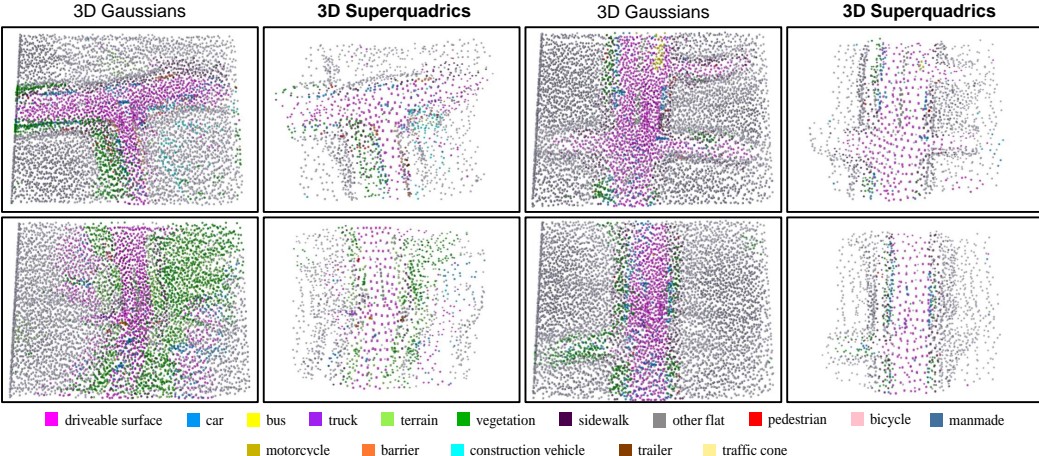

| 3D Gaussians | **3D Superquadrics** | 3D Gaussians | **3D Superquadrics** |

■ driveable surface ■ car ■ bus ■ truck ■ terrain ■ vegetation ■ sidewalk ■ other flat ■ pedestrian ■ bicycle ■ manmade

■ motorcycle ■ barrier ■ construction vehicle ■ trailer ■ traffic cone

Figure 8: **Visualizations of primitive position distributions learned by different methods.** Our approach produces well-structured spatial arrangements while using significantly fewer primitives.

## C   Additional Implementation Details

We provide implementation details of the prunning-and-splitting module. To clarify, we take the QuadricFormer with N superquadrics as an example and describe the process as follows:

**Initial Training**: We first train a QuadricFormer with B=4 quadric-encoder blocks and without the prunning-and-splitting module. The model starts from N randomly initialized superquadrics $Q_{init}$ and predicts adjusted superquadrics Q.

**Prunning-and-Splitting Module**: During experiments, we observed that some superquadrics in Q contribute little to scene modeling, which are usually located in empty regions with very small scales. To address this, we introduce the prunning-and-splitting module:

・ We divide all superquadrics in Q into two groups based on the product of their scales: the $N_{crop}$ superquadrics $C$ with the smallest scales and the remaining $N_{valid}$ superquadrics $V$, where $N = N_{valid} + N_{crop}$.

・ We discard the smaller superquadrics $C$ as they are most likely to contribute little to scene modeling.

・ We randomly sample $N_{crop}$ superquadrics from $V$ to form $S$. The features of $S$ remain unchanged, and only their positions are slightly adjusted.

**Further Refinement**: Finally, $S$ and $V$ are passed through two additional quadric-encoder blocks to further refine their attributes, resulting in the final superquadrics $Q_{final}$. At this stage, we load the pretrained model parameters and continue training for 10 more epochs.

