# OpenReview forum: "QuadricFormer: Scene as Superquadrics for 3D Semantic Occupancy Prediction"
_NeurIPS.cc/2025/Conference — NeurIPS 2025 poster_

### Official Review · Reviewer_VvnD · 2025-06-13

**Clarity:** 4
**Significance:** 3
**Originality:** 3
**Rating:** 5
**Confidence:** 5

**Summary:**

This paper introduces a novel method for 3D occupancy prediction that leverages superquadrics as a more efficient and expressive scene representation compared to traditional voxels or Gaussians. The core motivation is that the greater representative power of superquadrics allows them to model fine-grained scene geometry with fewer primitives, thereby reducing computational requirements. The authors propose a probabilistic superquadric mixture model, where each superquadric defines an occupancy probability distribution. This is integrated into a query-based network, QuadricFormer, which features a pruning-and-splitting module for efficient prediction. The method reports state-of-the-art (SOTA) results on the nuScenes dataset while demonstrating superior computational efficiency.

**Questions:**

I already included all questions in Strengths And Weaknesses section.

Overall, while this paper introduces a compelling and novel approach by leveraging superquadrics for efficient 3D occupancy prediction, its central claim -- that the observed gains stem directly from the superior representational power of superquadrics -- is not yet fully substantiated. The primary concern is that the performance and efficiency improvements may originate from the QuadricFormer architecture and its aggressive pruning module rather than the choice of primitive itself, a suspicion reinforced by qualitative results (Figs. 1 & 4) where the advertised "deforming" capability of superquadrics is not apparent. They often appear as simple, large-scale cuboids rather than adaptively capturing fine-grained geometric details. To resolve this, the authors should replace superquadrics with Gaussians within the same architectural framework is essential. The authors should provide new visualizations that offer close-up, side-by-side comparisons of a geometrically complex object (e.g., a vehicle, a traffic sign, or a building corner) as reconstructed by your method versus a Gaussian-based method. This direct comparison would make any superior detail-capturing ability immediately apparent. Finally, adjust the table to have fairer SOTA comparisons and evaluate on an additional dataset like SSCBench as in GaussianFormer2.

**Ethical Concerns:**

["NO or VERY MINOR ethics concerns only"]

**Final Justification:**

The authors have convincingly addressed all my concerns, and in my view, have also adequately responded to those raised by other reviewers.

This work introduces the use of superquadrics as a novel representation for 3D occupancy prediction in urban environments, while prior work only focused only on indoor scenes. The application to complex urban settings is both non-trivial and novel. The response clearly demonstrates that the observed performance gains are attributable to the use of superquadrics rather than auxiliary architectural components such as the pruning-and-splitting module. Notably, the method does not require depth initialization, which is a significant practical advantage.

The proposed approach achieves performance on par or better than recent SOTA models such as GaussianFormer-2, while operating under fewer constraints (i.e., no depth init). The improvements on large object classes (e.g., road surfaces) are particularly compelling, as shown in Tab. 1 on nuScenes. As the authors note, and I agree, modeling large flat surfaces with Gaussians is inefficient, whereas superquadrics (e.g., cuboids) offer a much more compact and interpretable representation.

Regarding benchmark selection, I find the evaluation strategy reasonable. The authors follow the standards established by recent leading works (e.g., TPVFormer, SurroundOcc, GaussianFormer). Given the concurrent release of Occ-3D and SurroundOcc, I do not find sufficient justification for requiring both benchmarks, especially when one is already widely accepted in the community.

The clarifications on the pruning-and-splitting module are now clear, and any needed refinements appear minor.

Finally, I disagree with reviewer #uPb2’s claim that the idea lacks impact. Demonstrating that superquadrics can be effectively applied to urban 3D occupancy prediction, with competitive results and practical benefits, is a meaningful contribution. Research impact should not be measured only by numerical performance but also by the novelty, practicality, and potential for future extensions.

In conclusion, the paper presents a novel idea, strong empirical evidence, and sound motivation. I recommend acceptance.

**Limitations:**

yes

**Paper Formatting Concerns:**

No concern.

**Quality:**

3

**Strengths And Weaknesses:**

# Strengths

- The paper is well-written and effectively illustrated. Figures 1, 2, and 3 are particularly helpful for building intuition, clarifying the differences between primitives, and explaining the method's architecture.

- While superquadrics have been explored for object-centric and indoor scene modeling, this paper presents a novel and timely application to the challenging domain of 3D occupancy prediction for autonomous driving.

- The central narrative -- that the superior representational capacity of superquadrics enables more efficient scene modeling by requiring fewer primitives -- is clearly articulated and consistently woven throughout the paper, making the motivation and design choices easy to follow.


# Weaknesses and Questions
- The paper's central argument is that superquadrics are inherently more efficient due to their expressive power. However, the evidence does not fully substantiate this claim. A key concern is that the observed reduction in primitives may stem more from the aggressive pruning-and-splitting module rather than the intrinsic properties of superquadrics. It is unclear if a similar architecture using Gaussians with the same pruning strategy would also achieve a comparable reduction in primitives.

    - Suggestion: To isolate the benefit of superquadrics, please replace the superquadrics in QuadricFormer with Gaussians, keeping the rest of the architecture (including the pruning module, query design, and training strategy) as identical as possible. This would directly test whether the representational choice itself is the primary driver of the efficiency gains.

- While the qualitative results in Figures 1 and 4 are visually appealing, they do not strongly demonstrate the claimed advantage of superquadrics in capturing fine-grained geometric details. The rendered superquadrics often appear as simple cuboids, quite similar in shape to what Gaussians would produce, just at a larger scale. The "deforming" capability that gives superquadrics their power (e.g., creating boxier or more cylindrical shapes) is not apparent. They do not seem to wrap around object corners or capture complex surfaces any better than their Gaussian counterparts in the visualizations provided. This weakens the core motivation of the paper.
    - Suggestion: Please provide more detailed visualizations that highlight the advantages of superquadrics in capturing fine-grained geometric details.

- The evaluation is currently limited to the nuScenes dataset. To demonstrate the generalizability and robustness of the proposed method, it is crucial to include results on other standard benchmarks for this task.

    - Suggestion: Please add results on SemanticKITTI and/or SSCBench-KITTI-360. Given that recent works like GaussianFormer-2 also report on SSCBench.

- Table 2 presents a confusing picture regarding efficiency. A superquadric contains more parameters than a Gaussian (e.g., shape parameters in addition to pose and scale). Therefore, it is counter-intuitive that a model with N superquadrics would be faster and require less memory than a model with N Gaussians, as the table seems to imply.

    - Suggestion: Please clarify this by augmenting Table 2. It would be very helpful to include columns that distinguish between network parameters and primitive parameters, and to explicitly state the number of primitives used in each configuration being compared. This would help readers understand the true source of the efficiency gains.

- The comparison with GaussianFormer-2 appears to be against a weaker variant (Ch=128, mIoU=20.02) rather than its best-performing published model (Ch=192, mIoU=20.82), which has a higher mIoU than the proposed method.

    - Suggestion: For a fair and transparent comparison, the paper should include the strongest published version of prior works.


- Several key hyperparameters are missing. Please specify:
    - The initial number of quadrics before the pruning-and-splitting stage.
    - The value of B, the number of blocks in the decoder (line 209). It is also unclear what a "block" refers to in this context.
    - The definition of "small-scale" and "large-scale" quadrics used for the pruning criteria (line 207).

- Recent relevant LiDAR-based methods are missing: PaSCo (CVPR'24), SCPNet (CVPR'23), and CAL (ICML'25).

---

> ### Author Rebuttal · Authors · 2025-07-31
>
> We thank the reviewer for the constructive comments and positive feedback on our paper. Regarding the concerns of the reviewer GNa2, we provide the following responses.
>
> > **W&Q1: A key concern is that the observed reduction in primitives may stem more from the aggressive pruning-and-splitting module rather than the intrinsic properties of superquadrics.**
>
> - Thanks for your valuable suggestion! To directly evaluate the advantage of superquadrics, we performed an ablation experiment where we replaced the superquadrics in QuadricFormer (with 1,600 primitives) with Gaussians, while keeping all other aspects of the model exactly the same.
>
> - The results in the table below show that the Gaussian-based model achieves much lower performance. This is because, without a heavy depth backbone and with a limited number of primitives, the Gaussian-based method cannot model the scene as effectively as superquadrics. In contrast, the superquadric-based approach maintains strong performance and efficiency.
>
> - These results demonstrate that the core improvements in performance and efficiency stem from the intrinsic representational power of superquadrics, rather than from architectural choices or the pruning strategy alone. We have included these findings in the revised manuscript for greater clarity. Thank you for prompting us to further validate this important point.
>
> |       Method       | Depth-Initialization | Pruning & Splitting | Latency (ms) | Memory (MB) |  IoU  | mIoU  |
> | :----------------: | :------------------: | :-----------------: | :----------: | :---------: | :---: | :---: |
> |   Gaussian-based   |          ✔           |          ✘          |     341      |    3075     | 28.99 | 18.73 |
> |   Gaussian-based   |          ✘           |          ✔          |     162      |    2554     | 29.31 | 18.91 |
> | Superquadric-based |          ✘           |          ✔          |     162      |    2554     | 30.71 | 20.04 |
>
> > **W&Q2: The rendered superquadrics often appear as simple cuboids, quite similar in shape to what Gaussians would produce, just at a larger scale. They do not seem to wrap around object corners or capture complex surfaces any better than their Gaussian counterparts in the visualizations provided.**
>
> - Thanks for your insightful suggestion. We acknowledge that superquadrics cannot capture precise object boundaries or highly complex surfaces in the occupancy prediction task. This is because the task uses 0.5-meter voxel labels, which only represent approximate object geometry rather than detailed surfaces.
>
> - However, superquadrics do possess clear advantages over Gaussians in modeling typical scene structures. For example, modeling large flat regions (such as road surfaces) would require many overlapping Gaussian ellipsoids, whereas only a small number of cuboid superquadrics are needed.
>
> - While images cannot be included in the NIPS rebuttal, we have added more detailed visualizations in the revised manuscript to better demonstrate the structural advantages of superquadrics in capturing geometric details within the limitations of current occupancy benchmarks.
>
>
> > **W&Q3: The evaluation is currently limited to the nuScenes dataset. To demonstrate the generalizability and robustness of the proposed method, it is crucial to include results on other standard benchmarks for this task.**
>
> - In response to the reviewer’s concerns about robustness and generalizability of our method, we have conducted additional experiments on the SSCBench-KITTI-360 [1] dataset. The results are summarized in the table below.
>
>
> - **Compared to previous dense voxel-based and sparse Gaussian-based methods, our method achieves comparable performance with much fewer scene primitives**. This highlights the efficiency of our method, as it can represent complex scenes with fewer expressive superquadrics without sacrificing much accuracy.
>
>
> - **We would like to clarify a key difference with GaussianFormer-2, which employs a heavy depth backbone to initialize Gaussians for improved performance**. While this strategy can yield better results, it comes at the cost of significantly reduced efficiency (i.e., high inference latency and memory usage), as evidenced by Table 2 in the paper. In contrast, to preserve the efficiency advantage of our method, we did not incorporate such a heavy depth-initialized module.
>
> - These results demonstrate both the effectiveness and the efficiency of our approach. We hope this addresses your concerns and further highlights the practical advantages of our method.
>
> - **Due to the limited time during the rebuttal period, we only conducted the experiments with 12800 superquadrics on SSCBench-KITTI-360. We will provide more experiments on Occ3D and SSCBench-KITTI-360 later**.
>
> |        Method        | Input Modality |   Number of Primitives   |  IoU  | mIoU  |
> | :------------------: | :------------: | :----------------------: | :---: | :---: |
> |     LMSCNet [2]      |     LiDAR      |            -             | 47.53 | 13.65 |
> |      SSCNet [3]      |     LiDAR      |            -             | 53.58 | 16.95 |
> |    MonoScene [4]     |     Camera     |   262144  (128x128x16)   | 37.87 | 12.31 |
> |    Voxformer [5]     |     Camera     |   262144  (128x128x16)   | 38.76 | 11.91 |
> |    TPVFormer [6]     |     Camera     | 81920 (256x256+256x32x2) | 40.22 | 13.64 |
> |    OccFormer [7]     |     Camera     |   262144  (128x128x16)   | 40.27 | 13.81 |
> |  GaussianFormer [8]  |     Camera     |          38400           | 35.38 | 12.92 |
> | GaussianFormer-2 [9] |     Camera     |          38400           | 38.37 | 13.90 |
> |       **Ours**       |     Camera     |        **12800**         | 36.81 | 12.86 |
>
> > **W&Q4: Table 2 presents a confusing picture regarding efficiency. It is counter-intuitive that a model with N superquadrics would be faster and require less memory than a model with N Gaussians, as the table seems to imply.**
>
> - The key reason for the efficiency difference is not the number of parameters per primitive, but rather the architectural choices. Specifically, **GaussianFormer-2 relies on a heavy depth backbone (ResNet-101) to initialize the Gaussians, which becomes the main bottleneck for inference speed and memory usage**. In contrast, QuadricFormer adopts cost-free random initialization for the superquadrics, which substantially reduces the computational and memory overhead.
>
> - We appreciate your suggestion, which helped us improve Table 2 to avoid any confusion. In the revised table, we have now indicated whether depth-backbone-based initialization is used and provided model parameters, making it clear where the efficiency gains originate.
>
>
> > **W&Q5: The comparison with GaussianFormer-2 appears to be against a weaker variant (Ch=128, mIoU=20.02) rather than its best-performing published model (Ch=192, mIoU=20.82), which has a higher mIoU than the proposed method.**
>
> - Thanks for your suggestion. **In Table 1, we only reported the results of GaussianFormer-2 (Ch=128, mIoU=20.02) with a feature dimension of 128 to ensure a fair comparison, since our method is under the same setting**. However, we agree that the best results should also be included to provide a more complete picture. In the revised version, we have added the strongest results of GaussianFormer-2 (Ch=192, mIoU=20.82) and clarified this in the manuscript.
>
> - Notably, even when compared with the best configuration of GaussianFormer-2, our QuadricFormer (with 12,800 superquadrics) still achieves superior performance with clear efficiency advantages. We appreciate your suggestion, which has improved the completeness and transparency of our experimental comparisons.
>
>
> > **W&Q6: Several key hyperparameters are missing.**
>
> Thanks for your detailed comments and suggestions. We address your questions as follows:
>
> 1. **Number of Quadrics before Pruning-and-Splitting:** The number of superquadrics remains unchanged before and after the pruning-and-splitting stage, since each pruned superquadric is replaced by a newly added one.
> 2. **Definition of a Block and Value of B in the Decoder:** A "quadric-encoder block" refers to a module that consists of a sparse convolutional layer, a cross-attention layer, and a residual update layer. Our model employs $B=4$ blocks to encode superquadrics.
> 3. **Small-scale and Large-scale Quadrics:** The distinction between small-scale and large-scale superquadrics is determined by the product of their scale values. More details can be found in our general response (GR3).
>
> We appreciate your valuable suggestions. We will include these clarifications in the revised version of our manuscript.
>
> > **W&Q7: Recent relevant LiDAR-based methods are missing: PaSCo (CVPR'24), SCPNet (CVPR'23), and CAL (ICML'25).**
>
> - Thanks for your suggestion. We will include these works and discuss them in the revised version of our paper.
>
> **References**
>
> [1] SSCBench: A Large-Scale 3D Semantic Scene Completion Benchmark for Autonomous Driving, IROS 2024.
>
> [2] LMSCNet: Lightweight Multiscale 3D Semantic Completion, 3DV 2020.
>
> [3] Semantic Scene Completion From a Single Depth Image, CVPR 2017.
>
> [4] MonoScene: Monocular 3D Semantic Scene Completion, CVPR 2022.
>
> [5] VoxFormer: Sparse Voxel Transformer for Camera-Based 3D Semantic Scene Completion, CVPR 2023.
>
> [6] Tri-Perspective View for Vision-Based 3D Semantic Occupancy Prediction, CVPR 2023.
>
> [7] OccFormer: Dual-path Transformer for Vision-based 3D Semantic Occupancy Prediction, ICCV 2023.
>
> [8] GaussianFormer: Scene as Gaussians for Vision-Based 3D Semantic Occupancy Prediction, ECCV 2024.
>
> [9] GaussianFormer-2: Probabilistic Gaussian Superposition for Efficient 3D Occupancy Prediction, CVPR 2025.

---

> ### Comment · Reviewer_VvnD · 2025-08-01
> **My initial response**
>
> Thanks the authors for their detailed response and the considerable effort put into conducting experiments, especially within such a short timeframe.
>
> ### Regarding W&Q1: Is reduction in primitives stem from the pruning-and-splitting module?
>
> I am convinced by the authors' response that the performance gains stem from the use of superquadrics rather than network architecture components like the "aggressive pruning-and-splitting module." Additionally, the fact that superquadrics don't require depth initialization is a significant advantage of the method.
>
> ### Regarding W&Q2: The rendered superquadrics appear as simple cuboids, similar to Gaussians?
>
> I agree that "modeling large flat regions (such as road surfaces) would require many overlapping Gaussian ellipsoids, whereas only a small number of cuboid superquadrics are needed." The results in Table 2 convincingly show that QuadricFormer performs best for "large flat regions" classes. However, Figure 2a is somewhat misleading.
>
>
>
> ### Regarding W&Q3 (also raised by #N2Bk, #cJXu, #GNa2): Experiments are done only on NuScenes?
>
> The authors' addition of results on SSCBench-KITTI-360 addresses this concern adequately.
>
> ### Regarding W&Q5 (also shared by #N2Bk, #cJXu): Why not compare against the strongest GaussianFormer-2 variant?
>
> The authors state that "In Table 1, we only reported the results of GaussianFormer-2 (Ch=128, mIoU=20.02) with a feature dimension of 128 to ensure a fair comparison, since our method is under the same setting." Could you elaborate on what "the same setting" means here? Typically, we compare the best against the best. If GaussianFormer-2 has a more advantageous setting, why not place QuadricFormer in the same setting for comparison?
>
> ###  Clarification on pruning-and-splitting module
>
> If the number of superquadrics remains unchanged (since each pruned superquadric is replaced by a newly added one), isn't the name "Pruning-and-splitting" misleading?
>
> Thank you for the clarifications on "Definition of a Block and Value of B in the Decoder" and "Small-scale and Large-scale Quadrics," as well as confirming that relevant works will be included in the revised paper.
>
> ### My current stand
> I find the comments from other reviewers compelling:
> - SOTA claim issues (#cJXu, #GNa2, #N2Bk)
> - Insufficient experiments (#uPb2)
> - Confusing pruning-and-splitting module (#N2Bk)
>
> I would like to see these discussions resolved before making my final decision.
>
>
> However, I disagree with reviewer #uPb2's assertion that "The idea lacks sufficient impact." Research impact is difficult to judge and cannot be dismissed simply due to lower performance metrics. Demonstrating that an idea works and showing its benefits compared to counterparts, even if not SOTA, is already sufficient contribution.
>
> I currently remain overall positive about the paper.

---

> > ### Author Response · Authors · 2025-08-04
> > **Response to Reviewer VvnD**
> >
> > We sincerely thank the reviewer for the timely response. We continue our discussions below.
> >
> > > **Regarding W&Q1:**
> >
> > Thank you for recognizing the effectiveness of superquadrics in our approach.
> >
> > > **Regarding W&Q2:**
> >
> > We agree with your observation. Figure 2a is intended as an illustrative diagram to highlight the advantages of superquadrics over Gaussians in geometry modeling. However, we acknowledge it may be somewhat misleading, as the occupancy prediction task does not require such fine-grained shapes. We will revise Figure 2a to better reflect the task while demonstrating the strengths of superquadrics.
> >
> > > **Regarding W&Q3:**
> >
> > Thanks for recognizing our results.
> >
> > > **Regarding W&Q5:**
> >
> > By “the same setting”, we mean QuadricFormer uses the same feature dimension of 128 as GaussianFormer-2 (Ch=128, mIoU=20.02). We did not use this higher feature dimension due to its impact on the speed and memory usage shown below. For a best-vs-best comparison, we are running additional experiments with QuadricFormer at 192 dimensions to match the strongest setting of GaussianFormer-2. These are still in progress, and we will update the results once available.
> >
> > |          Method           | Dimension of Features | Number of Primitives |  IoU  | mIoU  | Latency (ms) | Memory (MB) |
> > | :-----------------------: | :-------------------: | :------------------: | :---: | :---: | :----------: | :---------: |
> > | GaussianFormer-2 (Ch=128) |          128          |        12800         | 30.56 | 20.02 |     451      |    4535     |
> > | GaussianFormer-2 (Ch=192) |          192          |        12800         | 31.74 | 20.82 |     482      |    5839     |
> > |         **Ours**          |          128          |         1600         | 31.22 | 20.12 |     162      |    2556     |
> >
> > > **Regarding the pruning-and-splitting module:**
> >
> > The purpose of the pruning-and-splitting module is to enhance modeling accuracy rather than reduce the number of primitives. We remove small-scale superquadrics from empty regions that contribute little to scene modeling, and split large-scale ones in non-empty regions to achieve finer scene representation. We agree that "pruning-and-splitting" may be misleading. Since the core objective is to adjust the spatial distribution of superquadrics, "primitive relocating module" is a more appropriate name.
> >
> > For the other reviewers' comments, we now provide a short summary of our responses:
> >
> > > **SOTA claim issues (#cJXu, #GNa2, #N2Bk):**
> >
> > Sorry for the confusion. We have included a SOTA version (**with 12800 superquadrics**) in Table 2 but not in Table 1 (only reported results **with 1600 superquadrics**). We will fix Table 1 in the revised version.
> >
> > > **Insufficient experiments (#uPb2):**
> >
> > We did not report results on Occ3D because we want to fairly compare our method with the most related counterparts, i.e., the GaussianFormer series, so we followed their setting and employed SurroundOcc as the main benchmark. We have provided the additional results on SSCBench-KITTI-360 (with updated results shown below). We hope these results are sufficient to demonstrate the benefits of using superquadrics as primitives.
> >
> > **We are pleased to share the new results on SSCBench-KITTI-360**. By adjusting the hyperparameters (the loss weights), our method achieves an mIoU of 13.63 and IoU of 38.89 with 12800 superquadrics and without the depth-initialized module. The updated results are shown below:
> >
> > |                Method                | Input Modality | Number of Primitives |  IoU  | mIoU  |
> > | :----------------------------------: | :------------: | :------------------: | :---: | :---: |
> > |            GaussianFormer            |     Camera     |        38400         | 35.38 | 12.92 |
> > | GaussianFormer-2 (Depth Initialized) |     Camera     |        38400         | 38.37 | 13.90 |
> > |               **Ours**               |     Camera     |        12800         | 38.89 | 13.63 |
> >
> > > **Confusing pruning-and-splitting module (#N2Bk):**
> >
> > We summarize the implementation and effects of the pruning-and-splitting module.
> >
> > - The module improves scene modeling accuracy by re-distributing the superquadrics, rather than reducing their total number. After initial training, we remove superquadrics with small scales (often in empty regions) and replace them by splitting and relocating those from non-empty regions. Further refinement is achieved with extra quadric-encoder blocks.
> >
> > - Quantitative experiments with 1600 superquadrics show that this module consistently improves performance with minimal impact on inference latency and memory usage. Excessive pruning reduces the number of valid superquadrics and leads to performance drops. For detailed experimental results, please see our reply to Reviewer N2Bk.
> >
> > We sincerely appreciate your insightful review. We hope the results we provide are sufficient to demonstrate the benefits. Please let us know if you require additional information. Thank you again for your time and constructive comments.

---

> ### Comment · Reviewer_VvnD · 2025-08-04
> **All concerns addressed, I support acceptance unless critical issues arise**
>
> Great results! Thank you for the detailed response. I truly appreciate the additional experiments you conducted, especially over the weekend.
>
> Overall, I find that all concerns, including those raised by other reviewers, have been addressed convincingly:
>
> - The use of superquadrics as a new representation for 3D occupancy prediction is both novel and non-trivial. While previously explored in indoor scenes, this is the first time they are applied to urban environments.
>
> - The advantages of this method are clear: it matches or even outperforms GaussianFormer-2 (very recent SOTA), while operating under fewer constraints (e.g., no need for depth initialization). The improvements on large object classes are particularly noteworthy and observed on nuScenes. Not sure about SCCBench-KITTI360, maybe the authors should report per-class performance on this dataset too, I hope it doesn't require much additional work?
>
> - Regarding benchmarks, I believe Occ-3D and SurroundOcc were released around the same time, roughly between the CVPR’23 and ICCV’23 deadlines. I'm not convinced that Occ-3D is “more widely adopted,” as suggested by reviewer #uPb2. Evaluating on one benchmark seems sufficient, especially since the authors follow the standard benchmarks used by TPVFormer, SurroundOcc, GaussianFormer, and GaussianFormer-2, arguably the most influential works in this domain.
>
> - Finally, the pruning-and-splitting module is now much clearer, and I believe it can be revised with minimal effort.
>
> In summary, I’m convinced by the method’s merits and would support accepting the paper (will raise score to 5), unless other reviewers raise any critical points.

---

> > ### Author Response · Authors · 2025-08-09
> >
> > Thanks for your time and effort.
> > Your insightful review and valuable suggestions have significantly improved the clarity and quality of our work.

---

### Official Review · Reviewer_uPb2 · 2025-06-20

**Clarity:** 3
**Significance:** 2
**Originality:** 3
**Rating:** 3
**Confidence:** 4

**Summary:**

The paper QuadricFormer proposes a novel and efficient object-centric scene representation for 3D semantic occupancy prediction using superquadrics instead of traditional dense voxels or Gaussian primitives. By leveraging the geometric flexibility of superquadrics, the proposed QuadricFormer framework efficiently models complex scene structures with significantly fewer primitives. The paper introduces a probabilistic superquadric mixture model and design a pruning-and-splitting module to further enhance representation efficiency. Extensive experiments on the nuScenes dataset demonstrate that QuadricFormer achieves state-of-the-art performance while substantially reducing computational cost and memory usage.

**Questions:**

The main concerns, as outlined in the Weaknesses section, can be summarized as follows:

1. The proposed idea lacks sufficient impact.

2. The efficiency comparison requires further clarification.

3. The experiments are insufficient in terms of benchmark diversity and baseline comparisons.

4. The evaluation metrics are inadequate and need to be updated.

**Ethical Concerns:**

["NO or VERY MINOR ethics concerns only"]

**Final Justification:**

The authors' rebuttal partially addresses my concerns. However, the proposed method demonstrates only moderate performance on the SSCBench-KITTI-360 benchmark. Since the introduction of the Occ3D benchmark, it has become a widely adopted standard, with over a hundred methods evaluated on it. Additionally, SparseOcc introduces RayIoU, a more principled metric that extends the conventional mIoU. Therefore, comparison experiments with both the Occ3D benchmark and RayIoU are crucial for a comprehensive evaluation. The current experimental results remain insufficient. Given that further experiments are still necessary to validate the soundness and effectiveness of the proposed approach, I maintain my negative rating.

**Limitations:**

Yes

**Quality:**

2

**Strengths And Weaknesses:**

Strengths:
1. This paper proposes a novel and efficient object-centric scene representation for 3D semantic occupancy prediction based on superquadrics.
2. The proposed model achieves superior performance while significantly reducing computational cost and memory usage compared to previous methods.

Weaknesses:
1. **The idea lacks sufficient impact**.

While this paper provides a valuable attempt to apply superquadrics to the occupancy prediction task, the performance improvement over previous methods, such as GaussianFormer-2, appears to be limited.

Regarding efficiency, memory usage is influenced by various factors, including the backbone architecture, the number of layers, and other design choices. Therefore, the reported efficiency improvements are difficult to attribute solely to the use of superquadric representations. Based on the current information, the comparison in Table 2 does not appear to be entirely fair.

Overall, this paper seems to have made a modest contribution and, at best, would likely receive only a lukewarm response within the community.

2. **Insufficient experiments**.

The experiments are conducted only on the nuScenes dataset and primarily compared with outdated methods such as TPVFormer and SurroundOcc. Several recent state-of-the-art approaches, including SparseOcc [1], FB-Occ [2], CVT-Occ [3], and many others, are notably missing from the comparisons. Additionally, evaluations on more widely adopted benchmarks, such as Occ3D-nuScenes and Occ3D-Waymo [4], are essential to convincingly demonstrate the advantages of the proposed method.

Furthermore, SparseOcc [1] has challenged the use of IoU and mIoU as evaluation metrics, advocating for the adoption of RayIoU instead. An update of the evaluation metrics is therefore necessary to ensure fair and meaningful comparisons.

**Reference**:

[1] Fully Sparse 3D Occupancy Prediction, ECCV 2024.

[2] FB-OCC: 3D Occupancy Prediction based on Forward-Backward View Transformation, arXiv 2023.

[3] CVT-Occ: Cost Volume Temporal Fusion for 3D Occupancy Prediction, ECCV 2024.

[4] Occ3D: A Large-Scale 3D Occupancy Prediction Benchmark for Autonomous Driving, arXiv 2023.

---

> ### Author Rebuttal · Authors · 2025-07-31
>
> We thank the reviewer for the constructive comments. Regarding the concerns of the reviewer uPb2, we provide the following responses.
>
> > **W1 & Q1 & Q2: The proposed idea lacks sufficient impact. The efficiency comparison requires further clarification.**
>
> - **On the fairness and validity of efficiency comparison:**
>   1. We would like to clarify that our efficiency comparison is fair and meaningful. As noted, model efficiency (inference speed and memory usage) can be affected by various factors. To ensure a fair comparison with Gaussian-based methods, we made the model architectures and evaluation settings as consistent as possible.
>   1. **The only difference is that GaussianFormer-2 requires an additional depth backbone for Gaussian initialization, while our method dose not. Instead, we employ a lightweight pruning-and-splitting module to enhance performance.**
>   1. **Since the depth backbone in GaussianFormer-2 is a heavy ResNet-101, it becomes the main bottleneck for inference speed and memory usage.** As a result, even with the same number of primitives, our method achieves better performance and efficiency. This highlights the effectiveness and efficiency of our method.
>
>
> - **On performance gains and broader impact:**
>   1. We respectfully disagree that the performance improvements over GaussianFormer-2 are limited. While GaussianFormer-2 achieves strong results by leveraging a heavy depth backbone and increasing feature dimensions, this comes at the cost of significant efficiency loss (see Table 2).
>   1. **Without relying on these additional tricks, QuadricFormer (with 12800 superquadrics) surpasses GaussianFormer-2 in mIoU (21.11 vs 20.82) with superior efficiency (179 ms latency vs 451 ms)**. This demonstrates that our method not only exceeds the performance of existing baselines but also sets a new standard for efficiency, which is valuable for real-world applications.
>
> - In summary, our results demonstrate that the proposed superquadric representations leads to meaningful advances in both performance and efficiency, supporting the impact and relevance of our work. We thank you for your suggestions, which have encouraged us to further clarify these points in the revised manuscript.
>
> > **W2 & Q3 & Q4: The experiments are insufficient in terms of benchmark diversity and baseline comparisons. The evaluation metrics are inadequate and need to be updated.**
>
> - **Experiments on additional benchmarks:**
>
>   - Thanks for your valuable suggestion. Due to the limited time of the rebuttal period, we were only able to provide additional results on SSCBench-KITTI-360 [1], following the experimental protocol of GaussianFormer-2. The results are summarized in the table below.
>
>
>   - **Compared to previous dense voxel-based and sparse Gaussian-based methods, our method achieves comparable performance with much fewer scene primitives**. This highlights our method’s efficiency, as it is able to represent complex scenes with fewer expressive superquadrics without sacrificing much accuracy.
>
>
>   - **We would like to clarify a key difference with GaussianFormer-2, which employs a heavy depth backbone to initialize Gaussians for improved performance**. While this strategy can yield better results, it comes at the cost of significantly reduced efficiency (i.e., high inference latency and memory usage), as evidenced by Table 2 in the paper. In contrast, to preserve the efficiency advantage of our method, we did not incorporate such a heavy depth-initialized module.
>
>   - These results demonstrate both the effectiveness and the efficiency of our approach. We hope this addresses your concerns and further highlights the practical advantages of our method.
>
>   - **Due to the limited time during the rebuttal period, we only conducted the experiments with 12800 superquadrics on SSCBench-KITTI-360. We will provide more experiments on Occ3D and SSCBench-KITTI-360 later**.
>
>   |        Method        | Input Modality |   Number of Primitives   |  IoU  | mIoU  |
>   | :------------------: | :------------: | :----------------------: | :---: | :---: |
>   |     LMSCNet [2]      |     LiDAR      |            -             | 47.53 | 13.65 |
>   |      SSCNet [3]      |     LiDAR      |            -             | 53.58 | 16.95 |
>   |    MonoScene [4]     |     Camera     |   262144  (128x128x16)   | 37.87 | 12.31 |
>   |    Voxformer [5]     |     Camera     |   262144  (128x128x16)   | 38.76 | 11.91 |
>   |    TPVFormer [6]     |     Camera     | 81920 (256x256+256x32x2) | 40.22 | 13.64 |
>   |    OccFormer [7]     |     Camera     |   262144  (128x128x16)   | 40.27 | 13.81 |
>   |  GaussianFormer [8]  |     Camera     |          38400           | 35.38 | 12.92 |
>   | GaussianFormer-2 [9] |     Camera     |          38400           | 38.37 | 13.90 |
>   |       **Ours**       |     Camera     |        **12800**         | 36.81 | 12.86 |
>
> - **Evaluation metrics:**
>   - While SparseOcc [10] introduced the RayIoU metric, the majority of recent and leading works on the SSCBench-KITTI-360 continue to report results primarily with mIoU and IoU as evaluation metrics. Therefore, for consistency and comparability with prior art, we continue to evaluate our model using these established metrics.
>   - Nevertheless, we acknowledge the value of more comprehensive evaluation. In future work, we will conduct additional experiments on the Occ3D dataset and adopt metrics such as RayIoU to enable a more thorough comparison with more recent occupancy prediction methods.
>
> **References**
>
> [1] SSCBench: A Large-Scale 3D Semantic Scene Completion Benchmark for Autonomous Driving, IROS 2024.
>
> [2] LMSCNet: Lightweight Multiscale 3D Semantic Completion, 3DV 2020.
>
> [3] Semantic Scene Completion From a Single Depth Image, CVPR 2017.
>
> [4] MonoScene: Monocular 3D Semantic Scene Completion, CVPR 2022.
>
> [5] VoxFormer: Sparse Voxel Transformer for Camera-Based 3D Semantic Scene Completion, CVPR 2023.
>
> [6] Tri-Perspective View for Vision-Based 3D Semantic Occupancy Prediction, CVPR 2023.
>
> [7] OccFormer: Dual-path Transformer for Vision-based 3D Semantic Occupancy Prediction, ICCV 2023.
>
> [8] GaussianFormer: Scene as Gaussians for Vision-Based 3D Semantic Occupancy Prediction, ECCV 2024.
>
> [9] GaussianFormer-2: Probabilistic Gaussian Superposition for Efficient 3D Occupancy Prediction, CVPR 2025.
>
> [10] Fully Sparse 3D Occupancy Prediction, ECCV 2024.

---

> > ### Author Response · Authors · 2025-08-04
> > **New Results on SSCBench-KITTI-360**
> >
> > **We are pleased to share the new results on SSCBench-KITTI-360**. Due to time constraints, our initial response only included results from a single run, which was not fully optimized. After tuning the hyperparameters (specifically, the loss weights), our method achieves an mIoU of 13.63 and IoU of 38.89 with 12,800 superquadrics and without the depth-initialized module. The updated results are shown below:
> >
> > |                Method                | Input Modality | Number of Primitives |  IoU  | mIoU  |
> > | :----------------------------------: | :------------: | :------------------: | :---: | :---: |
> > |            GaussianFormer            |     Camera     |        38400         | 35.38 | 12.92 |
> > | GaussianFormer-2 (Depth Initialized) |     Camera     |        38400         | 38.37 | 13.90 |
> > |               **Ours**               |     Camera     |        12800         | 38.89 | 13.63 |

---

> > ### Comment · Reviewer_uPb2 · 2025-08-05
> > **Response**
> >
> > The authors' rebuttal partially addresses my concerns. However, since the introduction of the Occ3D benchmark, it has become a widely adopted standard, with over a hundred methods evaluated on it. Moreover, SparseOcc introduces RayIoU, a more principled metric that extends the conventional mIoU. The current comparisons overlook a large number of existing published methods, which significantly limits the completeness of the experimental evaluation. These aspects should be incorporated into the revised paper to strengthen its contribution.

---

> > > ### Comment · Area_Chair_QjLm · 2025-08-06
> > > **clarification needed**
> > >
> > > Dear uPb2 (the authors are included in this one),
> > >
> > > You mentioned that the current comparisons ignore "a large number of existing published methods". Could you please provide a specific list of papers the authors should have compared to?
> > >
> > > Thank you
> > > AC

---

> > > > ### Comment · Reviewer_uPb2 · 2025-08-06
> > > > **Response to AC**
> > > >
> > > > Dear AC,
> > > >
> > > > Since the introduction of the Occ3D benchmark, it has been cited nearly 300 times, and numerous milestone methods have been evaluated on it. Therefore, comparison with these works (a small part of them are as below) is essential to meet the high standards of NeurIPS.
> > > >
> > > > - Sparseocc: Rethinking sparse latent representation for vision-based semantic occupancy prediction
> > > > - Panoocc: Unified occupancy representation for camera-based 3d panoptic segmentation
> > > > - Fb-occ: 3d occupancy prediction based on forward-backward view transformation
> > > > - Renderocc: Vision-centric 3d occupancy prediction with 2d rendering supervision
> > > > - Cotr: Compact occupancy transformer for vision-based 3d occupancy prediction
> > > > - Opus: occupancy prediction using a sparse set
> > > > - Octreeocc: Efficient and multi-granularity occupancy prediction using octree queries
> > > > - Flashocc: Fast and memory-efficient occupancy prediction via channel-to-height plugin
> > > > - Occsora: 4d occupancy generation models as world simulators for autonomous driving
> > > > - Occgen: Generative multi-modal 3d occupancy prediction for autonomous driving
> > > > - Cvt-occ: Cost volume temporal fusion for 3d occupancy prediction
> > > > - Fully sparse 3d occupancy prediction

---

> > > > ### Comment · Reviewer_uPb2 · 2025-08-06
> > > > **Response to AC**
> > > >
> > > > Dear AC,
> > > >
> > > > I maintain my original opinion primarily due to the limited experimental evaluation. However, I also acknowledge the positive contributions of this paper and the authors’ thorough rebuttal. Therefore, if both the AC and other reviewers lean toward acceptance, I would also support that decision.

---

> ### Author Response · Authors · 2025-08-06
> **Response**
>
> Thanks to the AC and the reviewer for their valuable comments and feedback.
>
> We would like to clarify that Occ-3D and SurroundOcc were released around the same time and provide additional 3D occupancy annotations to the nuScenes data. Subsequent methods usually chose one of them as the occupancy evaluation benchmark on nuScenes.
>
> Typically, voxel-based 3D occupancy prediction methods targeting improving the model architecture (e.g., SparseOcc, FB-Occ, and CVT-Occ) and occupancy generation methods (e.g., OccSora) usually adopt Occ-3D. On the other hand, methods targeting developing new 3D representations to replace the voxel representation (e.g., GaussianFormer [1], GaussianFormer-2 [2]) usually adopt SurroundOcc.
>
> We followed this convention, and as our main goal is to validate the effectiveness and advantage of superquadrics as a new scene representation, we adopted SurroundOcc to ensure fair comparisons with the most relevant counterparts (e.g., the Gaussian representation in GaussianFormer, GaussianFormer-2). This choice is also supported by Reviewer #VvnD ("the authors follow the standard benchmarks used by TPVFormer, SurroundOcc, GaussianFormer, and GaussianFormer-2, arguably the most influential works in this domain.").
>
> Therefore, even though we agree that additional evaluation on Occ3D is a valuable enhancement, we respectfully disagree that the lack of it is a fundamental problem. We are currently trying our best to adapt our method to Occ3D and will provide the results once we have them. Once again, we appreciate the time and effort the AC and reviewer devoted to reviewing and improving our work. We hope the above response can help address your concerns. We are happy to answer any additional questions you may have.
>
> **References**
>
> [1] GaussianFormer: Scene as Gaussians for Vision-Based 3D Semantic Occupancy Prediction, ECCV 2024.
>
> [2] GaussianFormer-2: Probabilistic Gaussian Superposition for Efficient 3D Occupancy Prediction, CVPR 2025.

---

> ### Comment · Reviewer_VvnD · 2025-08-07
>
> > Since the introduction of the Occ3D benchmark, it has been cited nearly 300 times, and numerous milestone methods have been evaluated on it. Therefore, comparison with these works (a small part of them are as below) is essential to meet the high standards of NeurIPS.
>
> I found this comment is a bit bias. Regarding adoption metrics, SurroundOcc acctually demonstrates higher adoption:
>
> - Citations: 323 (SurroundOcc) vs 289 (Occ3D)
> - GitHub stars: 937 (SurroundOcc) vs 486 (Occ3D)

---

### Official Review · Reviewer_GNa2 · 2025-06-25

**Clarity:** 3
**Significance:** 2
**Originality:** 3
**Rating:** 4
**Confidence:** 4

**Summary:**

The paper replace the Gaussian with superquadrics as scene primitives in GaussianFormer to overcome the ellipsoidal shape limitation of Gaussian-based methods By leveraging a probabilistic superquadric mixture model and a pruning-splitting module, it concentrates primitives in occupied regions for better efficiency. Experiments on nuScenes show QuadricFormer achieves good performance with fewer primitives.

**Questions:**

- Could the authors provide more details on the pruning-and-splitting module? According to Table 4, this component plays a significant role in the model’s performance. However, the current version of the paper lacks sufficient explanation regarding its design and implementation.
- [minor] In Table 1, the caption mentions “Ch”, but this notation does not appear anywhere in the table itself.
- [minor] The statements in lines 244–245 do not align with the results reported in Table 1. The authors should revise the text to accurately reflect the experimental outcomes.

**Ethical Concerns:**

["NO or VERY MINOR ethics concerns only"]

**Final Justification:**

After reading all the responses, I still have concerns on the benchmark used. The paper evaluates their method on the dataset provided by SurroundOcc rather than widely adpoted benchmarks such as Occ3D. Although this dataset seems to be chosen by several works such as SurroundOcc, GaussianFormer, and GaussianFormer-2, it appears that these works originate from authors and institutions with close ties. In my view, this phenomenon is detrimental to the community, as it prevents researchers from fairly assessing works in the field.

Overall, the quality of the paper is good; however, my opinion remains quite borderline.

**Limitations:**

Yes.

**Quality:**

2

**Strengths And Weaknesses:**

Strengths:
- The motivation to replace Gaussians with a more general and expressive representation is commendable.
- The presentation of the paper is clear.

Weaknesses:
- The experimental evaluation in the current version is insufficient. Most existing occupancy prediction methods report results on benchmarks such as OpenOccupancy or Occ3D, which facilitates fair and consistent comparison across approaches. However, the authors only evaluate their model on the dataset introduced by SurroundOcc, limiting the ability of the community to properly assess the model’s effectiveness. I strongly encourage the authors to include additional evaluations to compare with other sparse occupancy prediction methods.
- The performance claims regarding QuadricFormer are misleading. The authors state in the main results (Tab. 2) that their model achieves state-of-the-art (SOTA) performance. However:
  - In Table 2, QuadricFormer achieves 31.22 IoU and 20.12 mIoU, which are both lower than those of SurroundOcc (31.49 IoU, 20.30 mIoU) and OccFormer (31.39 IoU). Highlighting QuadricFormer as the best-performing model is potentially misleading to readers.
  - Additionally, the reported numbers for GaussianFormer-2 (30.56 IoU, 20.02 mIoU) are not its best-performing results. According to its original paper, the best configuration of GaussianFormer-2 achieves 31.74 IoU and 20.82 mIoU, which outperforms QuadricFormer. The authors should ensure fair and accurate comparisons by reporting the strongest results for all baselines.

---

> ### Author Rebuttal · Authors · 2025-07-31
>
> We thank the reviewer for the constructive comments and positive feedback on our paper. Regarding the concerns of the reviewer GNa2, we provide the following responses.
>
> > **W1: The experimental evaluation in the current version is insufficient. I strongly encourage the authors to include additional evaluations to compare with other sparse occupancy prediction methods.**
>
> - Thanks for your valuable suggestion. Due to the limited time of the rebuttal period, we were only able to provide additional results on SSCBench-KITTI-360 [1], following the experimental protocol of GaussianFormer-2. The results are summarized in the table below.
>
>
> - Compared to previous dense voxel-based and sparse Gaussian-based methods, our method achieves comparable performance with much fewer scene primitives. This highlights our method’s efficiency, as it is able to represent complex scenes with fewer expressive superquadrics without sacrificing much accuracy.
>
>
> - **We would like to clarify a key difference with GaussianFormer-2, which employs a heavy depth backbone to initialize Gaussians for improved performance**. While this strategy can yield better results, it comes at the cost of significantly reduced efficiency (i.e., high inference latency and memory usage), as evidenced by Table 2 in the paper. In contrast, to preserve the efficiency advantage of our method, we did not incorporate such a heavy depth-initialized module.
>
> - These results demonstrate both the effectiveness and the efficiency of our approach. We hope this addresses your concerns and further highlights the practical advantages of our method.
>
> - **Due to the limited time during the rebuttal period, we only conducted the experiments with 12800 superquadrics on SSCBench-KITTI-360. We will provide more experiments on Occ3D and SSCBench-KITTI-360 later**.
>
> |        Method        | Input Modality |   Number of Primitives   |  IoU  | mIoU  |
> | :------------------: | :------------: | :----------------------: | :---: | :---: |
> |     LMSCNet [2]      |     LiDAR      |            -             | 47.53 | 13.65 |
> |      SSCNet [3]      |     LiDAR      |            -             | 53.58 | 16.95 |
> |    MonoScene [4]     |     Camera     |   262144  (128x128x16)   | 37.87 | 12.31 |
> |    Voxformer [5]     |     Camera     |   262144  (128x128x16)   | 38.76 | 11.91 |
> |    TPVFormer [6]     |     Camera     | 81920 (256x256+256x32x2) | 40.22 | 13.64 |
> |    OccFormer [7]     |     Camera     |   262144  (128x128x16)   | 40.27 | 13.81 |
> |  GaussianFormer [8]  |     Camera     |          38400           | 35.38 | 12.92 |
> | GaussianFormer-2 [9] |     Camera     |          38400           | 38.37 | 13.90 |
> |       **Ours**       |     Camera     |        **12800**         | 36.81 | 12.86 |
>
> > **W2.1: QuadricFormer achieves lower IoU and mIoU than SurroundOcc and OccFormer. Highlighting QuadricFormer as the best-performing model is potentially misleading to readers.**
>
> - We apologize for the lack of clarity in Table 1, where we only reported the performance of QuadricFormer with **1600** superquadrics. This presentation may have caused confusion about our SOTA claim. **Our model actually can be configured to use different numbers of superquadrics, and using more superquadrics yields better accuracy with only a slight reduction in efficiency.** As shown in Table 2, with **6400** and **12800** superquadrics, QuadricFormer achieves mIoUs of **20.79** and **21.11** and IoUs of **31.89** and **32.13**, respectively, surpassing all previous methods and achieving SOTA results.
>
> - Moreover, unlike GaussianFormer-2, we do not use the heavy depth-initialized module, so our model maintains high efficiency even with 6400 or 12800 superquadrics (see Table 2).
>
> - To avoid confusion, we have updated Table 1 in the revised manuscript to clarify the SOTA claim and clearly indicate the results for different numbers of superquadrics.
>
> > **W2.2: The reported numbers for GaussianFormer-2 (30.56 IoU, 20.02 mIoU) are not its best-performing results.**
>
> - Thanks for your suggestion. **In Table 1, we only reported the results of GaussianFormer-2 (Ch=128, mIoU=20.02) with a feature dimension of 128 to ensure a fair comparison, since our method is under the same setting**. However, we agree that the best results should also be included to provide a more complete picture. In the revised version, we have added the strongest results of GaussianFormer-2 (Ch=192, mIoU=20.82) and clarified this in the manuscript.
>
>
> - Notably, even when compared with the best configuration of GaussianFormer-2, our QuadricFormer (with 12,800 superquadrics) still achieves superior performance with clear efficiency advantages. We appreciate your suggestion, which has improved the completeness and transparency of our experimental comparisons.
>
>
> > **Q1: Could the authors provide more details on the pruning-and-splitting module?**
>
> We provide implementation details of the prunning-and-splitting module. To clarify, we take the QuadricFormer with $N$ superquadrics as an example and describe the process as follows:
>
> 1. **Initial Training:** We first train a QuadricFormer with $B=4$ quadric-encoder blocks and without the prunning-and-splitting module. The model starts from $N$ randomly initialized superquadrics $\mathbf{Q}_{init}$ and predicts adjusted superquadrics $\mathbf{Q}$.
>
> 2. **Prunning-and-Splitting Module:**
>    During experiments, we observed that some superquadrics in $\mathbf{Q}$ contribute little to scene modeling, which are usually located in empty regions with very small scales. To address this, we introduce the prunning-and-splitting module:
>    - We divide all superquadrics in $\mathbf{Q}$ into two groups based on the product of their scales: the $N_{crop}$ superquadrics $\mathbf{C}$ with the smallest scales and the remaining $N_{valid}$ superquadrics $\mathbf{V}$, where $N=N_{crop}+N_{valid}$.
>    - We discard the smaller superquadrics $\mathbf{C}$ as they are most likely to contribute little to scene modeling.
>    - We randomly sample $N_{crop}$ superquadrics from $\mathbf{V}$ to form $\mathbf{S}$. The features of $\mathbf{S}$ remain unchanged, and only their positions are slightly adjusted.
>
> 3. **Further Refinement:**
>    Finally, $\mathbf{S}$ and $\mathbf{V}$ are passed through two additional quadric-encoder blocks to further refine their attributes, resulting in the final superquadrics $\mathbf{Q}_{final}$. At this stage, we load the pretrained model parameters and continue training for 10 more epochs.
>
> > **M1 & M2**
>
> 1. Thanks for pointing out this editing error. We have now removed the incorrect "Ch" notation from the caption of Table 1.
> 2. We acknowledge that there was a mismatch between the text and results in Table 1. This was because Table 1 did not include model performance under settings of more than 1600 superquadrics, which were included in Table 2. We have updated Table 1 and revised the corresponding statements to ensure full consistency between our description and the reported results.
>
> **References**
>
> [1] SSCBench: A Large-Scale 3D Semantic Scene Completion Benchmark for Autonomous Driving, IROS 2024.
>
> [2] LMSCNet: Lightweight Multiscale 3D Semantic Completion, 3DV 2020.
>
> [3] Semantic Scene Completion From a Single Depth Image, CVPR 2017.
>
> [4] MonoScene: Monocular 3D Semantic Scene Completion, CVPR 2022.
>
> [5] VoxFormer: Sparse Voxel Transformer for Camera-Based 3D Semantic Scene Completion, CVPR 2023.
>
> [6] Tri-Perspective View for Vision-Based 3D Semantic Occupancy Prediction, CVPR 2023.
>
> [7] OccFormer: Dual-path Transformer for Vision-based 3D Semantic Occupancy Prediction, ICCV 2023.
>
> [8] GaussianFormer: Scene as Gaussians for Vision-Based 3D Semantic Occupancy Prediction, ECCV 2024.
>
> [9] GaussianFormer-2: Probabilistic Gaussian Superposition for Efficient 3D Occupancy Prediction, CVPR 2025.

---

> > ### Author Response · Authors · 2025-08-04
> > **New Results on SSCBench-KITTI-360**
> >
> > **We are pleased to share the new results on SSCBench-KITTI-360**. Due to time constraints, our initial response only included results from a single run, which was not fully optimized. After tuning the hyperparameters (specifically, the loss weights), our method achieves an mIoU of 13.63 and IoU of 38.89 with 12,800 superquadrics and without the depth-initialized module. The updated results are shown below:
> >
> > |                Method                | Input Modality | Number of Primitives |  IoU  | mIoU  |
> > | :----------------------------------: | :------------: | :------------------: | :---: | :---: |
> > |            GaussianFormer            |     Camera     |        38400         | 35.38 | 12.92 |
> > | GaussianFormer-2 (Depth Initialized) |     Camera     |        38400         | 38.37 | 13.90 |
> > |               **Ours**               |     Camera     |        12800         | 38.89 | 13.63 |

---

> > > ### Comment · Reviewer_GNa2 · 2025-08-07
> > >
> > > The authors' response and the updated experiments have mostly alleviated my concerns. I would like to to increase my rating. However, I believe the field's development would be hindered if works remain incomparable due to diverse evaluation benchmarks. For the community's benefit, I strongly recommend the authors to include results on Occ3D in the next version if possible.

---

> > > > ### Author Response · Authors · 2025-08-07
> > > > **Response to Reviewer GNa2**
> > > >
> > > > Thanks for your valuable suggestions and positive recognition of our work. We are currently trying our best to conduct experiments on the Occ3D. We will include the results in the updated version to enable a more comprehensive comparison with existing methods. The relevant code will also be released to benefit the community. Thank you again for your thoughtful feedback.

---

> ### Comment · Reviewer_VvnD · 2025-08-07
>
> I disagree with the reviewer. The claim that "works remain incomparable due to diverse evaluation benchmarks" is flawed. Diverse benchmarks actually enhance comparability by revealing method strengths and limitations across multiple perspectives.
>
> Each benchmark captures unique aspects of occupancy prediction - different sensor modalities, environmental conditions, and task formulations. Even benchmarks using the same dataset differ significantly: although Occ3D and SurroundOcc both use nuScenes, their approaches to aggregating 3D occupancy ground-truth are different. Furthermore, SurroundOcc open-sources their ground-truth generation pipeline while Occ3D doesn't provide theirs. This open-source approach not only ensures transparency and reproducibility but also enables other researchers to generate occupancy ground-truth for additional datasets, fostering broader community development rather than restricting progress to a single closed benchmark.
>
> Multiple benchmarks reveal method robustness and identify failure modes that single-benchmark evaluation would miss. The goal should be developing methods that perform well across diverse settings, not excelling on one standardized test.
>
> Of course, I agree that it would be more beneficial if the authors provide results on more benchmarks, however this also requires extensive computational resources that may not be available to all research groups

---

> > ### Author Response · Authors · 2025-08-09
> >
> > Thanks to the reviewer #VvnD for the constructive feedback and understanding. Due to time and computational resource constraints, we were unable to include results on Occ3D during the rebuttal period. In the revised version, we will include experiments on Occ3D to enable more comprehensive comparisons. We will also release our code to support reproducibility and benefit the community.

---

### Official Review · Reviewer_cJXu · 2025-06-26

**Clarity:** 3
**Significance:** 3
**Originality:** 3
**Rating:** 4
**Confidence:** 4

**Summary:**

This paper introduces a novel vision-based 3D semantic scene completion method that employs geometrically expressive superquadrics as scene primitives instead of traditional voxel-based representations. Extensive experiments on the nuScenes dataset demonstrate that this new representation enables a more lightweight 3D SSC framework with improved latency and memory efficiency, while maintaining competitive performance.

**Questions:**

- In Table 1, the authors claim to achieve state-of-the-art performance. However, both the IoU and mIoU scores are lower than those of existing methods. Please clarify this discrepancy.
- I would like to see results on additional datasets such as SemanticKITTI. This would be necessary to better demonstrate the generalizability and practical effectiveness of the proposed method.
- In Table 4, the best performance is achieved when the Crop & Split Number is set to 800. Would increasing this number further lead to better performance? Additionally, how does the Crop & Split Number affect inference speed and memory usage?

**Ethical Concerns:**

["NO or VERY MINOR ethics concerns only"]

**Final Justification:**

Thank you for your response. However, given the lower performance of the proposed method on the other dataset, I would like to maintain my initial score.

**Limitations:**

yes

**Paper Formatting Concerns:**

There are no formatting concerns.

**Quality:**

3

**Strengths And Weaknesses:**

Strengths
- The paper is well-written and the methodology is easy to understand with clear intuition.
- Applying superquadrics to 3D semantic scene completion is a novel, effective, and highly interesting approach.
- The experimental results are promising, particularly in terms of efficiency, and support the proposed approach.

Weaknesses
- The proposed method shows lower IoU and mIoU performance compared to SurrondOcc. However, this gap is acceptable considering the improved efficiency in terms of latency and memory usage.
- To further demonstrate the robustness of the proposed method, evaluations on more diverse datasets beyond a single benchmark, nuScene, would be beneficial.

---

> ### Author Rebuttal · Authors · 2025-07-31
>
> We thank the reviewer for the constructive comments and positive feedback on our paper. Regarding the concerns of the reviewer cJXu, we provide the following responses.
>
> > **Q1: In Table 1, the authors claim to achieve state-of-the-art performance. However, both the IoU and mIoU scores are lower than those of existing methods.**
>
> - We apologize for the lack of clarity in Table 1, where we only reported the performance of QuadricFormer with **1600** superquadrics. This presentation may have caused confusion about our SOTA claim. **Our model actually can be configured to use different numbers of superquadrics, and using more superquadrics yields better accuracy with only a slight reduction in efficiency.** As shown in Table 2, with **6400** and **12800** superquadrics, QuadricFormer achieves mIoUs of **20.79** and **21.11** and IoUs of **31.89** and **32.13**, respectively, surpassing all previous methods and achieving SOTA results.
>
> - Moreover, unlike GaussianFormer-2, we do not use the heavy depth-initialized module, so our model maintains high efficiency even with 6400 or 12800 superquadrics (see Table 2).
>
> - To avoid confusion, we have updated Table 1 in the revised manuscript to clarify the SOTA claim and clearly indicate the results for different numbers of superquadrics.
>
> > **Q2: I would like to see results on additional datasets such as SemanticKITTI. This would be necessary to better demonstrate the generalizability and practical effectiveness of the proposed method.**
>
> - In response to the reviewer’s concerns about robustness and generalizability of our method, we have conducted additional experiments on the SSCBench-KITTI-360 [1] dataset. The results are summarized in the table below.
>
>
> - **Compared to previous dense voxel-based and sparse Gaussian-based methods, our method achieves comparable performance with much fewer scene primitives**. This highlights our method’s efficiency, as it is able to represent complex scenes with fewer expressive superquadrics without sacrificing much accuracy.
>
>
> - **We would like to clarify a key difference with GaussianFormer-2, which employs a heavy depth backbone to initialize Gaussians for improved performance**. While this strategy can yield better results, it comes at the cost of significantly reduced efficiency (i.e., high inference latency and memory usage), as evidenced by Table 2 in the paper. In contrast, to preserve the efficiency advantage of our method, we did not incorporate such a heavy depth-initialized module.
>
> - These results demonstrate both the effectiveness and the efficiency of our approach. We hope this addresses your concerns and further highlights the practical advantages of our method.
>
> - **Due to the limited time during the rebuttal period, we only conducted the experiments with 12800 superquadrics on SSCBench-KITTI-360. We will provide more experiments on Occ3D and SSCBench-KITTI-360 later**.
>
>
> |        Method        | Input Modality |   Number of Primitives   |  IoU  | mIoU  |
> | :------------------: | :------------: | :----------------------: | :---: | :---: |
> |     LMSCNet [2]      |     LiDAR      |            -             | 47.53 | 13.65 |
> |      SSCNet [3]      |     LiDAR      |            -             | 53.58 | 16.95 |
> |    MonoScene [4]     |     Camera     |   262144  (128x128x16)   | 37.87 | 12.31 |
> |    Voxformer [5]     |     Camera     |   262144  (128x128x16)   | 38.76 | 11.91 |
> |    TPVFormer [6]     |     Camera     | 81920 (256x256+256x32x2) | 40.22 | 13.64 |
> |    OccFormer [7]     |     Camera     |   262144  (128x128x16)   | 40.27 | 13.81 |
> |  GaussianFormer [8]  |     Camera     |          38400           | 35.38 | 12.92 |
> | GaussianFormer-2 [9] |     Camera     |          38400           | 38.37 | 13.90 |
> |       **Ours**       |     Camera     |        **12800**         | 36.81 | 12.86 |
>
> > **Q3: Would increasing this number further lead to better performance? Additionally, how does the Crop & Split Number affect inference speed and memory usage?**
>
> - We provide more results to quantitatively analyze the impact of the pruning-and-splitting module on both model performance and efficiency. All experiments used a fixed total number of 1600 superquadrics.
>
> - **The results show that this module clearly improves performance with only a minor efficiency loss.** Moreover, changes in $N_{crop}$ have only a minor effect on inference speed and memory usage, allowing the model to achieve better performance while maintaining high efficiency.
>
> - The best performance is achieved when $N_{crop}=800$ (i.e., the crop & split number in the table). Increasing $N_{crop}$ beyond 800 actually leads to a slight drop in performance. We believe this is because excessive cropping removes too many superquadrics, including some valid ones, which negatively impacts the performance.
>
>   | Crop & Split Number | mIoU  |  IoU  | Latency (ms) | Memory (MB) |
>   | :-----------------: | :---: | :---: | :----------: | :---------: |
>   |          0          | 19.41 | 29.77 |     158      |    2554     |
>   |         200         | 19.65 | 30.35 |     161      |    2554     |
>   |         400         | 19.90 | 30.67 |     161      |    2554     |
>   |         800         | 20.12 | 31.22 |     162      |    2554     |
>   |        1200         | 19.91 | 30.41 |     164      |    2554     |
>
> **References**
>
> [1] SSCBench: A Large-Scale 3D Semantic Scene Completion Benchmark for Autonomous Driving, IROS 2024.
>
> [2] LMSCNet: Lightweight Multiscale 3D Semantic Completion, 3DV 2020.
>
> [3] Semantic Scene Completion From a Single Depth Image, CVPR 2017.
>
> [4] MonoScene: Monocular 3D Semantic Scene Completion, CVPR 2022.
>
> [5] VoxFormer: Sparse Voxel Transformer for Camera-Based 3D Semantic Scene Completion, CVPR 2023.
>
> [6] Tri-Perspective View for Vision-Based 3D Semantic Occupancy Prediction, CVPR 2023.
>
> [7] OccFormer: Dual-path Transformer for Vision-based 3D Semantic Occupancy Prediction, ICCV 2023.
>
> [8] GaussianFormer: Scene as Gaussians for Vision-Based 3D Semantic Occupancy Prediction, ECCV 2024.
>
> [9] GaussianFormer-2: Probabilistic Gaussian Superposition for Efficient 3D Occupancy Prediction, CVPR 2025.

---

> > ### Author Response · Authors · 2025-08-04
> > **New Results on SSCBench-KITTI-360**
> >
> > **We are pleased to share the new results on SSCBench-KITTI-360**. Due to time constraints, our initial response only included results from a single run, which was not fully optimized. After tuning the hyperparameters (specifically, the loss weights), our method achieves an mIoU of 13.63 and IoU of 38.89 with 12,800 superquadrics and without the depth-initialized module. The updated results are shown below:
> >
> > |                Method                | Input Modality | Number of Primitives |  IoU  | mIoU  |
> > | :----------------------------------: | :------------: | :------------------: | :---: | :---: |
> > |            GaussianFormer            |     Camera     |        38400         | 35.38 | 12.92 |
> > | GaussianFormer-2 (Depth Initialized) |     Camera     |        38400         | 38.37 | 13.90 |
> > |               **Ours**               |     Camera     |        12800         | 38.89 | 13.63 |

---

> > > ### Comment · Reviewer_cJXu · 2025-08-05
> > > **Response**
> > >
> > > Thank you for your thoughtful response. However, even considering the experimental results, a borderline acceptance seems to be the most appropriate evaluation. I wish you the best of luck with your work. Thank you.

---

### Official Review · Reviewer_N2Bk · 2025-07-02

**Clarity:** 3
**Significance:** 2
**Originality:** 2
**Rating:** 4
**Confidence:** 4

**Summary:**

This work proposes QuadricFormer, a novel 3D occupancy prediction model that uses probabilistic superquadrics instead of Gaussians to efficiently represent complex scene geometries. It achieves state-of-the-art performance on the nuScenes dataset with significantly improved representation efficiency.

**Questions:**

see weakness. If these concerns are clarified and the results remain solid, I would be happy to increase my score.

**Ethical Concerns:**

["NO or VERY MINOR ethics concerns only"]

**Final Justification:**

The author's clarification in the response addressed my concerns regarding the issues of SOTA comparison, generalization evaluation, and the design of the Pruning-and-Splitting Module. While this method has the potential to serve as a new baseline, the performance improvement is not particularly significant. Therefore, I consider this paper to be at the borderline of acceptance.

**Limitations:**

yes

**Paper Formatting Concerns:**

Not found.

**Quality:**

3

**Strengths And Weaknesses:**

**Strengths**:

1.The introduction of superquadrics for scene geometry representation is the core innovation of this paper. Compared to previous 3D Gaussian-based representations, superquadrics offer significantly greater geometric expressiveness, allowing complex structures—such as cuboids, cylinders, and planes—to be modeled with far fewer primitives.

2.The experimental results are impressive, particularly in terms of speed improvement, where the proposed method significantly reduces latency and memory consumption.

3.The paper is logically well-structured and clearly written, making it easy to follow and understand.

**Weaknesses**:

**1. Issue with SOTA Claim in Table 1**

While the paper claims state-of-the-art performance, the results in Table 1 do not fully support this assertion. Specifically:

OccFormer reports a higher IoU (31.39 vs. 31.22) than QuadricFormer.

SurroundOcc achieves both higher mIoU (20.30 vs. 20.12) and IoU (31.49 vs. 31.22).

Although the differences are relatively minor, the current claim of SOTA is not strictly accurate based on these metrics. The authors are encouraged to revise the statement to more precisely reflect the strengths of QuadricFormer—such as efficiency-performance trade-off or performance with fewer primitives—to ensure consistency between the claims and the presented results.

2.**The evaluation is limited to nuScenes.** Results on KITTI-360 or other datasets would help demonstrate the generalization and robustness of the proposed method.

3.**Pruning-and-Splitting Module Is Somewhat Confusing.**
The pruning-and-splitting module is described as a key innovation, yet its implementation is underspecified, and Table 4 contains a puzzlingly high IoU (39.77 for number=0) that contradicts other results. The efficiency gain attributed to this module is not clearly quantified. More clarity, quantitative impact, and validation are needed to justify this component.

---

> ### Author Rebuttal · Authors · 2025-07-31
>
> We thank the reviewer for the constructive comments and positive feedback on our paper. Regarding the concerns of the reviewer N2Bk, we provide the following responses.
>
> > **W1:  Issue with SOTA Claim in Table 1**
>
> - We apologize for the lack of clarity in Table 1, where we only reported the performance of QuadricFormer with **1600** superquadrics. This presentation may have caused confusion about our SOTA claim. **Our model actually can be configured to use different numbers of superquadrics, and using more superquadrics yields better accuracy with only a slight reduction in efficiency.** As shown in Table 2, with **6400** and **12800** superquadrics, QuadricFormer achieves mIoUs of **20.79** and **21.11** and IoUs of **31.89** and **32.13**, respectively, surpassing all previous methods and achieving SOTA results.
>
> - Moreover, unlike GaussianFormer-2, we do not use the heavy depth-initialized module, so our model maintains high efficiency even with 6400 or 12800 superquadrics (see Table 2).
>
>
> - To avoid confusion, we have updated Table 1 in the revised manuscript to clarify the SOTA claim and clearly indicate the results for different numbers of superquadrics.
>
> > **W2: The evaluation is limited to nuScenes.**
>
> - In response to the reviewer’s concerns about robustness and generalizability of our method, we have conducted additional experiments on the SSCBench-KITTI-360 [1] dataset. The results are summarized in the table below.
>
>
> - **Compared to previous dense voxel-based and sparse Gaussian-based methods, our method achieves comparable performance with much fewer scene primitives**. This highlights our method’s efficiency, as it is able to represent complex scenes with fewer expressive superquadrics without sacrificing much accuracy.
>
>
> - **We would like to clarify a key difference with GaussianFormer-2, which employs a heavy depth backbone to initialize Gaussians for improved performance**. While this strategy can yield better results, it comes at the cost of significantly reduced efficiency (i.e., high inference latency and memory usage), as evidenced by Table 2 in the paper. In contrast, to preserve the efficiency advantage of our method, we did not incorporate such a heavy depth-initialized module.
>
> - These results demonstrate both the effectiveness and the efficiency of our approach. We hope this addresses your concerns and further highlights the practical advantages of our method.
>
> - **Due to the limited time during the rebuttal period, we only conducted the experiments with 12800 superquadrics on SSCBench-KITTI-360. We will provide more experiments on Occ3D and SSCBench-KITTI-360 later**.
>
> |        Method        | Input Modality |   Number of Primitives   |  IoU  | mIoU  |
> | :------------------: | :------------: | :----------------------: | :---: | :---: |
> |     LMSCNet [2]      |     LiDAR      |            -             | 47.53 | 13.65 |
> |      SSCNet [3]      |     LiDAR      |            -             | 53.58 | 16.95 |
> |    MonoScene [4]     |     Camera     |   262144  (128x128x16)   | 37.87 | 12.31 |
> |    Voxformer [5]     |     Camera     |   262144  (128x128x16)   | 38.76 | 11.91 |
> |    TPVFormer [6]     |     Camera     | 81920 (256x256+256x32x2) | 40.22 | 13.64 |
> |    OccFormer [7]     |     Camera     |   262144  (128x128x16)   | 40.27 | 13.81 |
> |  GaussianFormer [8]  |     Camera     |          38400           | 35.38 | 12.92 |
> | GaussianFormer-2 [9] |     Camera     |          38400           | 38.37 | 13.90 |
> |       **Ours**       |     Camera     |        **12800**         | 36.81 | 12.86 |
>
> > **W3: Pruning-and-Splitting Module Is Somewhat Confusing.**
>
> - We provide implementation details of the prunning-and-splitting module. To clarify, we take the QuadricFormer with $N$ superquadrics as an example and describe the process as follows:
>
>   1. **Initial Training:** We first train a QuadricFormer with $B=4$ quadric-encoder blocks and without the prunning-and-splitting module. The model starts from $N$ randomly initialized superquadrics $\mathbf{Q}_{init}$ and predicts adjusted superquadrics $\mathbf{Q}$.
>
>   2. **Prunning-and-Splitting Module:**
>      During experiments, we observed that some superquadrics in $\mathbf{Q}$ contribute little to scene modeling, which are usually located in empty regions with very small scales. To address this, we introduce the prunning-and-splitting module:
>
>      - We divide all superquadrics in $\mathbf{Q}$ into two groups based on the product of their scales: the $N_{crop}$ superquadrics $\mathbf{C}$ with the smallest scales and the remaining $N_{valid}$ superquadrics $\mathbf{V}$, where $N=N_{crop}+N_{valid}$.
>      - We discard the smaller superquadrics $\mathbf{C}$ as they are most likely to contribute little to scene modeling.
>      - We randomly sample $N_{crop}$ superquadrics from $\mathbf{V}$ to form $\mathbf{S}$. The features of $\mathbf{S}$ remain unchanged, and only their positions are slightly adjusted.
>
>   3. **Further Refinement:**
>      Finally, $\mathbf{S}$ and $\mathbf{V}$ are passed through two additional quadric-encoder blocks to further refine their attributes, resulting in the final superquadrics $\mathbf{Q}_{final}$. At this stage, we load the pretrained model parameters and continue training for 10 more epochs.
>
> - We provide more results to quantitatively analyze the impact of the pruning-and-splitting module on both model performance and efficiency. All experiments used a fixed total number of 1600 superquadrics. **The results show that this module clearly improves performance with only a minor efficiency loss.** The best performance is achieved when $N_{crop}=800$ (i.e., the crop & split number in the table). Increasing $N_{crop}$ further actually reduces the number of valid superquadrics and leads to a drop in performance. Moreover, changes in $N_{crop}$ have only a minor effect on inference speed and memory usage, allowing the model to achieve better performance while maintaining high efficiency.
>
>   | Crop & Split Number | mIoU  |  IoU  | Latency (ms) | Memory (MB) |
>   | :-----------------: | :---: | :---: | :----------: | :---------: |
>   |          0          | 19.41 | 29.77 |     158      |    2554     |
>   |         200         | 19.65 | 30.35 |     161      |    2554     |
>   |         400         | 19.90 | 30.67 |     161      |    2554     |
>   |         800         | 20.12 | 31.22 |     162      |    2554     |
>   |        1200         | 19.91 | 30.41 |     164      |    2554     |
>
> - Regarding the puzzlingly high IoU (39.77 for number=0) in Table 4, we apologize for this typographical error. The correct value is 29.77 (not 39.77), and this typo has been fixed in the revised manuscript.
>
> **References**
>
> [1] SSCBench: A Large-Scale 3D Semantic Scene Completion Benchmark for Autonomous Driving, IROS 2024.
>
> [2] LMSCNet: Lightweight Multiscale 3D Semantic Completion, 3DV 2020.
>
> [3] Semantic Scene Completion From a Single Depth Image, CVPR 2017.
>
> [4] MonoScene: Monocular 3D Semantic Scene Completion, CVPR 2022.
>
> [5] VoxFormer: Sparse Voxel Transformer for Camera-Based 3D Semantic Scene Completion, CVPR 2023.
>
> [6] Tri-Perspective View for Vision-Based 3D Semantic Occupancy Prediction, CVPR 2023.
>
> [7] OccFormer: Dual-path Transformer for Vision-based 3D Semantic Occupancy Prediction, ICCV 2023.
>
> [8] GaussianFormer: Scene as Gaussians for Vision-Based 3D Semantic Occupancy Prediction, ECCV 2024.
>
> [9] GaussianFormer-2: Probabilistic Gaussian Superposition for Efficient 3D Occupancy Prediction, CVPR 2025.

---

> > ### Comment · Reviewer_N2Bk · 2025-08-04
> > **response**
> >
> > Thank you for your response. The authors have adequately addressed my concerns. I have decided to raise my score.

---

### Note · Authors · 2025-08-13

Dear PCs, SACs, ACs, and all of our reviewers,

We sincerely thank you for your thorough reviews and valuable feedback. Your comments and discussions have greatly helped us in improving our work, and we have answered and addressed most of the concerns.

We appreciate your recognition of our paper's strengths:

- The innovative use of superquadrics as scene primitives for 3D occupancy prediction, enabling greater geometric expressiveness and efficiency compared to voxel or Gaussian-based methods.
- Efficient scene modeling with significantly fewer primitives, resulting in substantial gains in computational efficiency and strong performance on the nuScenes benchmark.
- Clear motivation, well-structured writing, and intuitive illustrations that make the methodology easy to follow.

We have also addressed most of the concerns:

- Added results on SSCBench-KITTI-360 to demonstrate the generalizability and robustness of our method beyond the nuScenes dataset.
- Provided more details and results on the pruning-and-splitting module, including its design, quantitative impact, and ablation analysis.
- Conducted more ablations and discussions to validate the geometric advantages of superquadrics over Gaussians.
- Revised the main results to enable more comprehensive experimental comparisons and to better support our claim of state-of-the-art performance.

Regarding the Occ3D evaluation highlighted by reviewer #uPb2, we would like to clarify that existing peer-reviewed 3D occupancy prediction papers usually adopted one annotation system of Occ3D and SurroundOcc for evaluation on nuScenes. As our main goal is to validate the effectiveness and advantage of superquadrics as a new scene representation, we adopted SurroundOcc to ensure fair comparisons with the most relevant counterparts (e.g., the Gaussian representation in GaussianFormer, GaussianFormer-2). Therefore, we think the lack of Occ3D results is not a fundamental problem, but we will include Occ3D experiments in the revised version to benefit the community.

Finally, we thank all reviewers for recognizing our contribution. We believe that our work advances the community by **proposing a superquadric-based 3D scene representation to enhance geometric expressiveness and modeling efficiency for 3D occupancy prediction**. We will release code, data, and checkpoints to facilitate further research.

Thank you again for your time and efforts.

Best regards,

Authors of Submission 3377

---

### Decision · Program_Chairs · 2025-09-17

**Decision:**

Accept (poster)

**Comment:**

The paper introduces a method for 3D occupancy prediction using superquadrics as a scene geometry representation in urban environments. It demonstrates that superquadrics can model complex scene structures with significantly fewer primitives compared to traditional dense voxels or Gaussian primitives.

The paper received one accept, three borderline accepts, and one borderline reject. All reviewers noted that (a) the method is novel, and (b) the experiments show superior performance while substantially reducing computational cost and memory usage compared to prior methods. The main weakness, raised by the negative reviewer (uPb2), was the lack of exhaustive comparisons with other methods discussed during the author–reviewer exchange. In response, reviewer VvnD argued during the AC-reviewer discussion:
"While I agree with reviewer uPb2 that SOTA results on Occ3D would be valuable, their absence doesn't diminish this paper's scientific contribution. The main contribution is demonstrating superquadrics' benefits over Gaussians and voxels. Achieving SOTA often requires dataset-specific engineering tricks that obscure core methodological contributions. Additionally, I think requiring every paper to beat SOTA is bad as it shifts focus from fundamental insights to incremental optimizations."

The AC agrees with this perspective: the novelty of the method should be valued over incremental SOTA improvements. Accordingly, the AC recommends acceptance. Still, the authors are strongly encouraged to address uPb2’s concerns and carefully consider all reviewer feedback from both the reviews and the author-reviewer discussion when preparing the final version of the paper!